# Stronger Neyman Regret Guarantees for Adaptive Experimental Design

**Georgy Noarov** [1]  **Riccardo Fogliato** [2]  **Martin Bertran** [2]  **Aaron Roth** [1,2]

## Abstract

We study the design of adaptive, sequential experiments for unbiased average treatment effect (ATE) estimation in the design-based potential outcomes setting. Our goal is to develop adaptive designs offering *sublinear Neyman regret*, meaning their efficiency must approach that of the hindsight-optimal nonadaptive design. Recent work (Dai et al., 2023) introduced ClipOGD, the first method achieving $\widetilde{O}(\sqrt{T})$ expected Neyman regret under mild conditions. In this work, we propose adaptive designs with substantially stronger Neyman regret guarantees. In particular, we modify ClipOGD to obtain anytime $\widetilde{O}(\log T)$ Neyman regret under natural boundedness assumptions. Further, in the setting where experimental units have pretreatment covariates, we introduce and study a class of contextual "multigroup" Neyman regret guarantees: Given any set of possibly overlapping groups based on the covariates, the adaptive design outperforms each group's best non-adaptive designs. In particular, we develop a contextual adaptive design with $\widetilde{O}(\sqrt{T})$ anytime multigroup Neyman regret. We empirically validate the proposed designs through an array of experiments.

## 1. Introduction

Randomized control trials (RCTs) play a central role in a variety of settings where causal effects need to be accurately measured, spanning healthcare and epidemiology, policy-making, the social sciences, econometrics, e-commerce, and beyond. In the classic potential outcomes framework (Neyman, 1923; Rubin, 1974), a central estimand is the average treatment effect (ATE) – the average individual causal effect across experimental units. To obtain precise estimates of the ATE, we generally seek estimators that are unbiased and have low variance.

In many cases, RCTs are run sequentially: Experimental units arrive one by one, and each unit is assigned to treatment or control adaptively, based on previous outcomes or auxiliary information. The data-driven nature and flexibility of these experiments suggest that such adaptive trials can achieve substantial efficiency gains over standard fixed designs, as shown in domains ranging from political science (Offer-Westort et al., 2021; Blackwell et al., 2022) to medicine (Chow & Chang, 2008; Villar et al., 2015; FDA, 2019). However, so far adaptive experiments have received limited attention (Hu & Rosenberger, 2006) and have been rarely used in practice due to concerns that adaptivity could invalidate standard statistical guarantees (van der Laan, 2008). Indeed, classic solutions for improving estimator efficiency in the batch setting, such as Neyman allocation (Neyman, 1992), can be nontrivial to extend to the sequential setting.

Recently, a growing body of work (Hahn et al., 2011; Kato et al., 2020; Li & Owen, 2024; Dai et al., 2023; Cook et al., 2023) has made progress on this front by introducing multi-stage adaptive designs that estimate the ATE via inverse-probability weighting (IPW)-type estimators with adaptively adjusted propensity scores. [1] Our work contributes to this literature by developing novel adaptive sequential designs for IPW-based ATE estimation with efficiency guarantees. Crucially, our methods –unlike most existing work– are developed within the finite-population setting (Wager, 2024), where the ATE is defined as a deterministic function of the observed population rather than a superpopulation parameter. This distinction ensures robustness to treatment effect heterogeneity and temporal data drift, challenges that can undermine conventional superpopulation-based designs.

**Our Contributions**  We focus on the design of adaptive RCTs to estimate the ATE as efficiently as the best-in-hindsight IPW design from some benchmark class, up to

---

[1]Department of Computer and Information Science, University of Pennsylvania. [2]Amazon Web Services. Correspondence to: Georgy Noarov <gnoarov@seas.upenn.edu>.

*Proceedings of the 42$^{nd}$ International Conference on Machine Learning*, Vancouver, Canada. PMLR 267, 2025. Copyright 2025 by the author(s).

Georgy Noarov conducted part of this work as an intern at Amazon Web Services.

[1]In parallel, studies that fall into the multi-armed bandits literature have developed adaptive designs for finding *reward-maximizing* treatments (arms) or policies, which is a distinct, and conflicting, objective than estimation efficiency (Zhang et al., 2020; 2021; Hadad et al., 2021; Xu et al., 2016; 2024).

error terms. Specifically, we aim to minimize the *Neyman regret* (Kato et al., 2020; Dai et al., 2023) – a measure comparing the variance of our adaptive estimator to that of the variance-minimizing nonadaptive Bernoulli trial where units are treated with some fixed probability. Currently, to our knowledge Dai et al. (2023)'s ClipOGD method is the only adaptive design achieving sublinear Neyman regret in the finite-population setting. This method guarantees $\widetilde{O}(\sqrt{T})$ expected regret for any $T$-unit trial under moment-bounded potential outcomes. However, two important questions arise:

I. Can we develop designs with better regret rates? Dai et al. (2023) conjectured that $\widetilde{O}(\sqrt{T})$ is the minimax Neyman rate.

II. Can we develop context-aware designs that use pre-treatment covariates to improve efficiency?

In this work, we answer both these questions affirmatively:

**Contribution I: Exponentially Improved Noncontextual Neyman Regret Bound.** We show that, under a natural strengthening of Dai et al. (2023)'s assumptions on the outcomes, we can modify ClipOGD to attain an anytime-valid Neyman regret bound of $\widetilde{O}(\log T)$.[2] To achieve this speedup, we leverage the strong convexity of the Neyman objective under our stricter lower-bounding assumption on the outcomes, which as we show leads to near-logarithmic regret via techniques introduced by (Hazan et al., 2007). Moreover, it can be shown that even under the weaker outcome lower bound assumption of Dai et al. (2023), our adaptive design can be tweaked to have the asymptotic efficiency of $(1 + \epsilon) V^* + \widetilde{O}\left(\frac{\log T}{T}\right)$ for any $\epsilon > 0$, where $V^*$ denotes the optimal nonadaptive design variance; the interpretation is that any $(1 + \epsilon)$-multiplicative approximation to the optimal variance can be attained at this fast rate. We validate the greater efficiency of our proposed design against that of ClipOGD through a suite of experiments on synthetic and real-world data.

**Contribution II: Adaptive Designs with Contextual Neyman Regret Guarantees.** We next develop a novel adaptive design MGATE (Multi-Group ATE) that leverages pre-treatment covariates to improve efficiency relative to the non-contextual setting. In a nutshell, given an arbitrary predefined finite collection $\mathcal{G} \subseteq 2^{\mathcal{X}}$ of contextual groups defined by the covariates (e.g., demographics), we propose a no $\mathcal{G}$-multigroup-Neyman-regret adaptive design that obtains sublinear regret simultaneously on all subsequences of

---

[2]In fact, a lower bounding construction in the very recent work of Li et al. (2024) shows that the best possible Neyman regret is $\Omega(1)$ even in the more relaxed superpopulation setting — and so our method achieves a *best-of-both-worlds* guarantee, up to logarithmic factors.

experimental units corresponding to the groups in $\mathcal{G}$. Critically, we also allow for overlapping groups, i.e., units can simultaneously belong to multiple groups. A key challenge here is to balance the treatment probabilities in a way that balances the efficiency of the ATEs estimates across groups. Our proposed design leverages a variation of the "sleeping experts" approach (Blum & Lykouris, 2020; Acharya et al., 2024) used in the online learning literature (Lee et al., 2022; Deng et al., 2024), that deals with the limited feedback and the fact that the observed objective values do not live in an a-priori bounded range. The method achieves $\widetilde{O}(\sqrt{T})$ multigroup Neyman regret. We also empirically validate its performance.

Our multigroup guarantees can be interpreted through the lens of group ATE (GATE) estimation (Chernozhukov et al., 2017; Semenova & Chernozhukov, 2021; Zimmert & Lechner, 2019). GATE occupies a middle ground between ATE, which measures the average effect over the entire sequence, and CATE (conditional ATE), which measures the ATE conditionally on each covariate vector. Existing works on GATE, however, are mainly focused on learning data-driven disjoint groups to improve overall ATE estimation. In contrast, our objective is to simultaneously ensure efficient GATE inference for any family of arbitrarily overlapping groups. This is related in motivation (though distinct in technique) to the recent work of (Kern et al., 2024) who use "multiaccuracy" to make CATE inference robust to certain kinds of distribution shift.

We expect that such multigroup efficiency guarantees can be broadly useful, and hope future work will study multigroup adaptive designs beyond the sequential finite-population setting that we focus on in this paper.

For an additional discussion of related work, including relevant independent work in the superpopulation setting, please see Appendix A.

**Organization** In Section 2, we introduce our general setting and objectives. In Section 3, we focus on the (vanilla) non-contextual setting, and present and analyze our adaptive design ClipOGD$^{\mathrm{SC}}$, which achieves near-logarithmic Neyman regret. We prove the main regret bound in Theorem 3.2 and demonstrate further guarantees on the adaptive design.

In Section 4, we introduce the notion of multigroup Neyman regret, and present our multigroup adaptive design MGATE (Algorithm 2), which achieves $\widetilde{O}(\sqrt{T})$ multigroup Neyman regret as shown in Theorem 4.2. Furthermore, in Appendix D we provide a general multigroup design (Algorithm 7) that significantly generalizes MGATE. In Section 5, we compare the empirical performance of our adaptive designs to the Dai et al. (2023) ClipOGD design on an array of real-world and synthetic sequential experimental design tasks.

## 2. Preliminaries

**Setting** We work in the design-based, sequential variant of the potential outcomes setting (Neyman, 1923; Rubin, 1974; Imbens & Rubin, 2015). A finite number of experimental units in the population arrive one by one at rounds $t \in \mathbb{N}_+$. Each unit has two associated fixed potential outcomes, only one of which can be observed: treatment outcome $y_t(1) \in \mathbb{R}$ and control outcome $y_t(0) \in \mathbb{R}$.

In the basic setting, the observed outcome is the only information the experimenter receives about the units. A richer setting is one where before choosing treatment or control for unit $t$, the Experimenter is given access to *pre-treatment covariate* $x_t \in \mathcal{X}$, where $\mathcal{X}$ is a feature space of arbitrary nature (e.g. $\mathcal{X}$ may be a finite-dimensional vector space). In this paper, we will study both settings: the noncontextual setting in Section 3 and the contextual one in Section 4.

**Adaptive Design** In a randomized controlled trial (RCT), the experimenter (randomly) decides whether to apply treatment or control to each unit, and observes the corresponding outcome but not the counterfactual. These randomized decisions for all units constitute the experimental design. We study adaptive experimental designs, described as follows.

---

### $T$-round Adaptive Design Protocol

Potential outcomes $\{(y_t(1), y_t(0))\}_{t \in [T]}$ are generated upfront (but not shown to Experimenter). Then, sequentially for each unit $t = 1 \ldots T$:

1. (*Contextual* setting only) Experimenter observes pre-treatment covariate $x_t \in \mathcal{X}$.

2. Experimenter sets treatment probability $p_t$.

3. Experimenter flips bias-$p_t$ coin to obtain realized treatment decision: $Z_t \sim \text{Bernoulli}(p_t)$.

4. Experimenter observes outcome $Y_t = y_t(Z_t)$.

---

By contrast, the standard nonadaptive (Bernoulli) trial fixes upfront the same treatment probability $p_t = p$ for all units $t$, and uses it throughout the experiment without any adjustments.

Our estimand of interest is the average treatment effect (*ATE*), which corresponds to the difference between the average outcomes of treatment and control units in the population. We provide the formal definition below.

**Definition 2.1** (ATE). The *average treatment effect* for potential outcomes $\{(y_t(1), y_t(0))\}_{t=1}^T$ is:

$$\tau_T = \frac{1}{T} \sum_{t=1}^T y_t(1) - y_t(0).$$

A classical estimator of the ATE is the adaptive IPW estimator (Horvitz & Thompson, 1952), which employs inverse probability weighting. We define it next.

**Definition 2.2** (Adaptive IPW Estimator). The *adaptive IPW estimator* of the ATE $\tau_T$ is:

$$\hat{\tau}_T = \frac{1}{T} \sum_t Y_t \left( \frac{Z_t}{p_t} - \frac{1 - Z_t}{1 - p_t} \right).$$

This estimator is unbiased, meaning that for any outcomes $\{(y_t(0), y_t(1))\}_{t=1}^T$ and any adaptive design $(p_t)_{t=1}^T$ with all $p_t \in (0, 1)$, we have $\mathbb{E}[\hat{\tau}_T] = \tau_T$. Thus, no matter what adaptive design Experimenter employs, the induced adaptive IPW estimator will always be unbiased. However, the estimator's variance will vary based on the design, making some designs more efficient than others.

**Objective: Minimize Variance of ATE Estimator** Our main goal will be to construct adaptive designs that asymptotically approach the variance of the best-in-hindsight experimental design in some benchmark class. A basic class of designs is that of nonadaptive designs, parameterized by the choice of fixed propensity $p \in (0, 1)$. Formally, we measure the *Neyman regret* (Kato et al., 2020; Dai et al., 2023) of any proposed adaptive design as the (time-rescaled) difference between its IPW estimator variance and the variance of same estimator under the most efficient nonadaptive design.

To define Neyman regret, note (see Proposition 2.2 of Dai et al. (2023)) that $\text{Var}[\hat{\tau}_T] = \sum_{t=1}^T \mathbb{E}[f_t(p_t)] / T^2 - k_{\text{ATE}}$, where $f_t(p) := y_t(1)^2/p + y_t(0)^2/(1-p)$ is the variance of the propensity-$p$ IPW estimator at unit $t$, and $k_{\text{ATE}} = \sum_{t=1}^T (y_t(1) - y_t(0))^2 / T^2$ is a design-independent term. We are now ready to provide the formal definition.

**Definition 2.3** (Neyman Regret (Kato et al., 2020; Dai et al., 2023)). The Neyman regret of adaptive design $(p_t)_{t=1}^T$ on a potential outcomes sequence $\{(y_t(1), y_t(0))\}_{t=1}^T$ is:[3]

$$\text{RegVar}_T = \max_{p_T^* \in (0,1)} \sum_{t=1}^T f_t(p_t) - f_t(p_T^*).$$

Thus the variance of the IPW estimator for a design $(p_t)_{t=1}^T$ differs from that of the best nonadaptive design by exactly $\text{RegVar}_T / T^2$, justifying the Neyman regret definition.

Our goal will be to develop adaptive designs with sublinear expected Neyman regret: $\mathbb{E}[\text{RegVar}_T] = o(T)$, or equivalently with vanishing average expected Neyman regret: $\mathbb{E}[\text{RegVar}_T / T] = o(1)$. We call any design that satisfies this a no-regret design.

---

[3]"Var" stands for variance, as Neyman regret captures the rescaled estimator variance associated with the design.

# 3. Efficient Non-Contextual ATE Estimation

We now present our first contribution: An adaptive design that achieves $\widetilde{O}(\log T)$ Neyman regret under natural assumptions on the outcomes. We begin by discussing the $\widetilde{O}(\sqrt{T})$-Neyman regret design ClipOGD of Dai et al. (2023), and then modifying it to better exploit the strongly convex structure of the Neyman objective. Next, we discuss further guarantees on our method's performance.

## 3.1. Adaptive Design with Logarithmic Neyman Regret

**Meta-Design: ClipOGD**  The first finite-population design that achieves sublinear Neyman regret, ClipOGD, was introduced by Dai et al. (2023). Leveraging the fact that the per-round Neyman objectives $f_t(p)$ are convex in $p$, it performs a modified version of online gradient descent (OGD) on $f_t$ to adaptively modify the treatment probabilities $p_t$.

The complicating factor is that the gradients of $f_t$ diverge when $p$ is close to 0 or 1: standard OGD analyses typically require explicit or implicit bounds on the gradients of the objective (Hazan et al., 2016), so vanilla projected OGD on the entire interval $[0, 1]$ will not work without modification. ClipOGD solves this problem by clipping the OGD iterates $\{p_t\}_{t \in \mathbb{N}_+}$ to be within a nested family $\{[\delta_t, 1-\delta_t]\}_{t \in \mathbb{N}_+}$ of subintervals of $(0, 1)$, which gradually expand to cover the whole interval in the infinite time limit (i.e., $\lim_{t \to \infty} \delta_t = 0$). The expansion is needed to handle cases when $p_T^*$ is close to the boundary. In view of this, we let $\delta_t = 1/h(t)$ for all $t \in \mathbb{N}_+$, where $h : \mathbb{N}_+ \to \mathbb{R}_{>0}$ is some strictly increasing function with $\lim_{t \to \infty} h(t) = \infty$. We call $\delta_t$ the *clipping rate*, $h$ the *clipping function*, and refer to any adaptive design $(p_t)_{t \in \mathbb{N}_+}$ that satisfies $1/h(t) \le p_t \le 1 - 1/h(t)$ for all $t$ as $h$-clipped. Algorithm 1 gives the pseudocode for ClipOGD. Here, $\prod_S(x)$ denotes the projection of $x$ onto interval $S \subset (0, 1)$.

---

**Algorithm 1** ClipOGD  (Dai et al., 2023)

Initialize $p_0 \leftarrow 0.5$ and $g_0 \leftarrow 0$
**for** units $t = 1, 2, \ldots$ **do**
    Set step size $\eta_t > 0$ and clipping rate $\delta_t \in (0, 0.5)$
    Set treatment probability $p_t \leftarrow \prod_{[\delta_t, 1-\delta_t]} (p_{t-1} - \eta_t \cdot g_{t-1})$
    Set treatment decision $Z_t \sim \text{Bernoulli}(p_t)$
    Observe outcome $Y_t \leftarrow y_t(Z_t)$
    Set gradient estimate: $g_t \leftarrow Y_t^2 \left( -\frac{Z_t}{p_t^3} + \frac{1-Z_t}{(1-p_t)^3} \right)$
**end for**

---

**ClipOGD$^0$: A $\widetilde{O}(\sqrt{T})$ Regret Design**  In their paper, Dai et al. (2023) analyzed and provided guarantees for a specific instantiation of ClipOGD, where $\eta_t = \sqrt{1/T}$ and $\delta_t = 0.5 \cdot t^{-1/\alpha}$ where $\alpha = \sqrt{5 \log T}$ for all $t = 1, \ldots, T$. For clarity, we call this design ClipOGD$^0$. Their main result proves that

ClipOGD$^0$ has $\widetilde{O}(\sqrt{T})$ Neyman regret under a moment assumption on the outcomes: $0 < c \le (\frac{1}{T} \sum_{t=1}^{T} y_i(t)^2)^{1/2}$ and $(\frac{1}{T} \sum_{t=1}^{T} y_i(t)^4)^{1/4} \le C$ for $i \in \{0, 1\}$ and some $c \le C$. However, the learning rate of ClipOGD$^0$ has several drawbacks. First, it is too conservative, precluding improvement in Neyman regret beyond $\widetilde{O}(\sqrt{T})$. Second, it is horizon-dependent, making it necessary to know (or commit to) $T$ upfront. Finally, it is constant rather than decreasing, so the design probabilities will jump around (rather than gradually converge) during any given run of ClipOGD$^0$.

**ClipOGD$^{\text{SC}}$: Our $\widetilde{O}(\log T)$ Regret Design**  We now present an adaptive design called ClipOGD$^{\text{SC}}$ that addresses these issues: It uses the learning rate $\eta_t \sim 1/t$ that, under Assumption 3.1, (1) achieves an exponentially improved Neyman regret bound, (2) is *anytime*, i.e., does not require advance knowledge of the time horizon $T$, and (3) its propensities converge in $L_2$ to the hindsight-best propensity. Our Neyman regret bound relies on a stricter assumption than the one made by Dai et al. (2023)'s, which we detail below.

**Assumption 3.1** (Bounds on Potential Outcomes).
There exist positive constants $c, C$ such that outcomes $\{(y_t(0), y_t(1))\}_{t \ge 1}$ satisfy for all time horizons $T$:

$$\max_{t \ge 1}\{|y_t(0)|, |y_t(1)|\} \le C,$$

$$c \le \min_{t \ge 1} \left(y_t(0)^2 + y_t(1)^2\right)^{1/2}$$

$$c \le \min_{i \in \{0,1\}} \left(\frac{1}{T} \sum_{t=1}^{T} y_t(i)^2\right)^{1/2}.$$

Next, let $h_{\text{inv}}$ be the inverse function of $h$, defined via the identity $h_{\text{inv}} \circ h = h \circ h_{\text{inv}} = \text{Id}$. Our main result is the following Neyman regret bound in terms of $T$, $h$, and $h_{\text{inv}}$.

**Theorem 3.2** (Stronger Neyman Regret Bound). *Suppose Assumption 3.1 is satisfied with $C$, $c$ the corresponding constants. Let $h : \mathbb{N}_+ \to \mathbb{R}_{>0}$ be strictly increasing. Let ClipOGD$^{\text{SC}}$ be the adaptive design that instantiates Algorithm 1 with learning rate $\eta_t = 1/(2c^2 t)$ and clipping rate $\delta_t = 1/h(t)$. Then, ClipOGD$^{\text{SC}}$ attains the following anytime-valid Neyman regret bound:*

$$\mathbb{E}[\text{RegVar}_T] = O\left((h(T))^5 \cdot \log(T) + (h_{\text{inv}}(1 + C/c))^2\right). \tag{1}$$

*Since $h$ can be chosen to grow arbitrarily slowly, we can get:* $\mathbb{E}[\text{RegVar}_T] = \widetilde{O}(\log T)$.

The proof is contained in Appendix B. It exploits the strong convexity of the Neyman objectives $f_t$ enabled by Assumption 3.1 (hence the 'SC' in ClipOGD$^{\text{SC}}$), by applying

the techniques for analyzing strongly convex gradient descent (Hazan et al., 2007; Rakhlin et al., 2012).

Compared to the analysis in Dai et al. (2023), we make explicit the dependence of the regret of ClipOGD on the clipping rate. Note that the choice of $h$ is flexible in the sense that any $h(t) = o(t^{0.2-\varepsilon})$ for any $\varepsilon > 0$ will result in a regret bound that is sublinear in $T$. From a practical standpoint, however, picking $h$ may be a nontrivial affair, as a slower-growing $h$ will have a faster-growing inverse mapping $h_{\text{inv}}$. While the $h_{\text{inv}}$-dependent term in the regret bound is constant in $T$, it can still be large in the constants of the problem. Intuitively, if $C/c$ is large, the optimal propensity $p_T^*$ may be near the boundary and convergence may be slow. We hope future work will further explore the 'well-conditioning' properties of Neyman regret.

### 3.2. Convergence of Adaptive Treatment Probabilities

We now investigate the trajectory of treatment probabilities $(p_t)_{t \geq 1}$ produced by ClipOGD$^{\text{SC}}$. Ideally, these propensities would converge to the optimal probabilities $(p_T^*)_{T \geq 1}$ as $T$ grows large. By tweaking the arguments used in establishing our Neyman regret bounds of Theorem 3.2, we can obtain convergence in squared means (and hence in probability). The next claims formalize this result. In particular, we first establish a quantitative bound on the $L_2$ convergence of our propensities to the benchmark ones. (See Appendix B for the derivation.)

**Lemma 3.3** ($L_2$-Deviation from Benchmark Design). *The deviation of the design probabilities of ClipOGD$^{\text{SC}}$ from the best nonadaptive design probabilities is $L_2$-bounded for all $T$ as:*

$$\mathbb{E}\left[(p_T - p_T^*)^2\right] \leq -\Theta\left(\frac{\mathbb{E}[\text{RegVar}_T]}{T}\right) + O\left(\frac{(h(T))^2 \log T}{T}\right).$$

This implies the following $L_2$-convergence result, subject to an assumption on the Neyman regret of ClipOGD$^{\text{SC}}$ which asks for it to not consistently outperform the optimal nonadaptive design.

**Corollary 3.4** ($L_2$-Convergence to Benchmark Design). *Assume ClipOGD$^{\text{SC}}$ has asymptotically nonnegative Neyman regret:* $\liminf_{T \to \infty} \frac{\mathbb{E}[\text{RegVar}_T]}{T} \geq 0$. *Then, its propensities $(p_t)_{t \geq 1}$ will converge to the benchmark nonadaptive propensities $(p_T^*)_{T \geq 1}$ in squared means:* $\mathbb{E}\left[(p_T - p_T^*)^2\right] \to 0$ *as* $T \to \infty$.

In the special case of sequences of potential outcomes that are (i.i.d.) samples from a superpopulation, the regret nonnegativity holds automatically, implying that our adaptive design will necessarily converge to the best nonadaptive design without further assumptions.

**Corollary 3.5** (Convergence in the Superpopulation Setting). *Suppose that the outcomes are drawn i.i.d. from a*

superpopulation: $(y_t(0), y_t(1)) \sim \mathcal{D}$ *for all $t \geq 1$ and any fixed distribution $\mathcal{D}$. Then, ClipOGD$^{\text{SC}}$ guarantees that* $\mathbb{E}\left[(p_T - p^*)^2\right] \to 0$ *at the rate $\widetilde{O}(\log T/T)$, and thus in particular that $p_T \to p^*$ in probability.*

*Proof.* In the superpopulation setting, *any* adaptive design will have nonnegative Neyman regret: $f_t(p) = f(p) = \mathbb{E}[y(1)^2]/p + \mathbb{E}[y(0)^2]/(1-p)$ has the same optimum $p^* = \left(1 + \mathbb{E}[(y_t(0))^2]/\mathbb{E}[(y_t(1))^2]\right)^{-1}$ for all units $t$, so $\mathbb{E}[\text{RegVar}_T] = \mathbb{E}\left[\sum_{t=1}^{T}(f(p_t) - f(p^*))\right] \geq 0.$ ∎

### 3.3. Valid CIs for the Adaptive IPW Estimator

We now turn to the issue of endowing the IPW estimator $\hat{\tau}_T$ induced by our adaptive design with asymptotically valid confidence intervals (CIs). In general, the existence and construction of valid CIs for $\hat{\tau}_T$ delicately depends on the choice of the design. However, we will now see that a construction of Dai et al. (2023) lends conservative CIs to all $h$-clipped adaptive designs with vanishing regret.

To formalize this result, we make a standard assumption: that the outcome sequences are not perfectly anticorrelated. To state it, define "empirical second raw moments" of the two outcome populations as: $S_T(i)^2 := \frac{1}{T}\sum_{t=1}^{T}(y_t(i))^2$ for $i \in \{0, 1\}$.

**Assumption 3.6** (Correlation of Outcome Populations (Dai et al., 2023)). *For a constant $c_\rho > 0$ and all $T \geq 1$, the running correlation $\rho_T$ of the sequences $\{(y_t(0), y_t(1))\}_{t \geq 1}$ satisfies:*

$$\rho_T \geq -1 + c_\rho, \quad \text{where } \rho_T := \frac{\frac{1}{T}\sum_{t=1}^{T} y_t(1)y_t(0)}{S_T(1)S_T(0)}.$$

**Theorem 3.7** (CIs for Clipped Adaptive Designs). *Suppose the potential outcomes satisfy Assumption 3.1 and Assumption 3.6. Consider any $h$-clipped adaptive design $(p_t)_{t \geq 1}$ with vanishing Neyman regret:* $\lim_{T \to \infty} \text{RegVar}_T = 0$. *Let* $\text{VB} = \frac{4}{T}S_T(1)S_T(0)$ *be a conservative upper bound on the hindsight-best nonadaptive variance. Then, letting $(Z_t)_{t \geq 1}$ be the treatment decisions, the estimator of Dai et al. (2023) given by:*

$$\widehat{\text{VB}} = \frac{4}{T}\sqrt{\left(\frac{1}{T}\sum_{t=1}^{T}(y_t(1))^2\frac{Z_t}{p_t}\right)\left(\frac{1}{T}\sum_{t=1}^{T}(y_t(0))^2\frac{1-Z_t}{1-p_t}\right)}$$

*converges to* VB *in probability at rate $O_p\left(\sqrt{h(T)/T}\right)$.*

*Consequently, $\widehat{\text{VB}}$ can be used to construct asymptotically valid Chebyshev-type confidence intervals for the adaptive IPW estimator $\hat{\tau}_T$ under any adaptive design satisfying the above conditions. Specifically, for any confidence level $\alpha \in (0, 1]$:*

$$\liminf_{T \to \infty} \Pr\left[\tau_T \in \left[\hat{\tau}_T \pm \alpha^{-1/2}\sqrt{\widehat{\text{VB}}}\right]\right] \geq 1 - \alpha.$$

The proof for Theorem 3.7 is outlined in Appendix C.

# 4. Efficient Multigroup ATE Estimation

**The Contextual Setting** Section 3 covers non-contextual adaptive designs that only observe outcomes. A contextual adaptive design, however, also observes pre-treatment covariates $x_t \in \mathcal{X}$ at the start of each round, which can help predict potential outcomes $(y_t(0), y_t(1))$. We can leverage this extra information to improve treatment assignments and outcome estimation.

**A Multigroup Formulation** We frame the contextual setting in a multigroup way. Before the experiment, we have a finite set of context-defined groups $\mathcal{G} = \{G_1, G_2, \dots\}$, each $G \subseteq \mathcal{X}$, where $\mathcal{X}$ is the feature space. Any covariate vector $x_t$ can belong to none, one, or more groups. The group definition is dependent on the specifics of the task, e.g., in a medical application the features $x_t$ could represent a patient's health history.

Our objective in a multigroup setting, informally, is to design an adaptive scheme that offers ATE estimation efficiency guarantees (such as Neyman regret guarantees) not only on average over the entire sequence of units but also on each subsequence that results from conditioning on units belonging to a group $G$, simultaneously for all groups $G \in \mathcal{G}$.

## 4.1. A New Metric: Multigroup Neyman Regret

We introduce multigroup Neyman regret as a strengthening of (vanilla) Neyman regret. Specifically, given any contextual group collection $\mathcal{G}$, $\mathcal{G}$-multigroup Neyman regret is the maximum Neyman regret that an adaptive design achieves over any group $G$ in the collection. We formalize it next.

**Definition 4.1** ($\mathcal{G}$-Multigroup Neyman Regret)**.** Given any group collection $\mathcal{G} \subseteq 2^{\mathcal{X}}$, the group-conditional Neyman regret of an adaptive design $\mathcal{A}$ on any group $G \in \mathcal{G}$ is defined as:

$$\text{RegVar}_T(\mathcal{A}; G)$$
$$:= \mathbb{E}\left[ \max_{p^* \in (0,1)} \sum_{t=1}^{T} \mathbb{1}[x_t \in G] \left( f_t(p_t) - f_t(p^*) \right) \right].$$

The $\mathcal{G}$-multigroup Neyman regret of $\mathcal{A}$ is then defined as its maximum group-conditional Neyman regret over all groups $G \in \mathcal{G}$:

$$\text{RegVarMG}_T(\mathcal{A}; \mathcal{G}) := \max_{G \in \mathcal{G}} \text{RegVar}_T(\mathcal{A}; G).$$

## 4.2. Achieving $\widetilde{O}(\sqrt{T})$ Multigroup Neyman Regret

We now present in Algorithm 2 an adaptive design which we call MGATE (for Multi-Group ATE) and achieves the $\widetilde{O}(\sqrt{T})$ multigroup Neyman regret bound.

---

**Algorithm 2** $\mathcal{A}_{MGATE}$: Multigroup Adaptive Design

Receive clipping function $h : \mathbb{N}_+ \to \mathbb{R}_{>0}$
Receive number of groups $d = |\mathcal{G}|$
Set group counts $n_0 \leftarrow \mathbf{0}^d$
Initialize $p_1 \leftarrow 0.5 \cdot \mathbf{1}^d$ // At round $t$, $p_t = (p_{t,G})_{G \in \mathcal{G}}$ will contain group propensities
Initialize $w'_1 \leftarrow \mathbf{1}^d, L_0 \leftarrow \mathbf{0}^d, q_0 \leftarrow 0$ // Parameters used to update group weights
**for** $t = 1, 2, \dots$ **do**
  Receive covariate vector $x_t \in \mathcal{X}$, determine the set of active groups $\mathcal{G}_t = \{G : x_t \in G, G \in \mathcal{G}\}$
  Cast $\mathcal{G}_t$ as indicator vector $a_t \in \{0,1\}^d$ ($a_{t,G} = 1 \iff G \in \mathcal{G}_t$). Set group counts: $n_t \leftarrow n_{t-1} + a_t$
  Normalize group weights: $w_{t,\text{eff}} \leftarrow \frac{a_t \odot w'_t}{\langle a_t, w'_t \rangle}$ // Set inactive group weights to $0$
  Set effective treatment probability: $p_{t,\text{eff}} \leftarrow \langle w_{t,\text{eff}}, p_t \rangle$ // Aggregate group propensities
  Set treatment decision: $Z_t \sim \text{Bernoulli}(p_{t,\text{eff}})$
  Receive realized outcome: $Y_t \leftarrow y_t(Z_t)$
  **for** active groups $G \in \mathcal{G}_t$ **do**
    /* Update group propensities using group-specific ClipOGD$^{\text{SC}}$-type update */
    Set estimated Neyman gradient as:
    $\widetilde{g}_{t,G} \leftarrow Y_t^2 \left( \frac{Z_t}{p_{t,\text{eff}}} + \frac{1-Z_t}{1-p_{t,\text{eff}}} \right) \left( -\frac{Z_t}{p_{t,G}^2} + \frac{1-Z_t}{(1-p_{t,G})^2} \right)$
    Update $p_{t+1,G} \leftarrow \underset{[\delta_{t,G}, 1-\delta_{t,G}]}{\Pi} (p_{t,G} - \eta_{t,G} \cdot \widetilde{g}_{t,G})$,
    where $\eta_{t,G} \leftarrow \frac{1}{2c^2 \cdot n_{t,G}}$ and $\delta_{t,G} \leftarrow \frac{1}{h(n_{t,G})}$
    /* Get losses used to update group weights */
    Set estimated Neyman loss as:
    $\widetilde{\ell}_{t,G} \leftarrow Y_t^2 \left( \frac{Z_t}{p_{t,\text{eff}}} + \frac{1-Z_t}{1-p_{t,\text{eff}}} \right) \left( \frac{Z_t}{p_{t,G}} + \frac{1-Z_t}{1-p_{t,G}} \right)$
  **end for**
  **for** inactive groups $G \notin \mathcal{G}_t$ **do**
    Set $p_{t+1,G} \leftarrow p_{t,G}$ and $\widetilde{\ell}_{t,G} \leftarrow 0$ // Inactive groups are not updated
  **end for**
  /* Update group weights: Higher cumulative group losses $\to$ larger weights */
  Set surrogate loss: $\ell_t \leftarrow a_t \odot \left( \widetilde{\ell}_t - \langle \widetilde{\ell}_t, w_{t,\text{eff}} \rangle \right)$
  Set $L_t \leftarrow L_{t-1} + \ell_t$ and $q_t \leftarrow q_{t-1} + \|\ell_t\|_2^2$
  Update group weights: $w'_{t+1} \leftarrow \max \left\{ \mathbf{0}^d, -\frac{1}{\sqrt{q_t}} L_t \right\}$
**end for**

---

**Additional Notation:** We use $\odot$ to denote elementwise

vector multiplication, and let $\mathbf{1}^d, \mathbf{0}^d$ be $d$-dimensional all-ones and all-zeros vectors. Also note that the update of $w'_{t+1}$ takes an *elementwise* maximum of the vectors, and assumes that $0/0 = 0$ to account for the corner case $q_t = 0$.

**Algorithm Description:** Given a collection $\mathcal{G}$ of $d$ groups, in each round MGATE reads off the currently active groups $\mathcal{G}_t \subseteq \mathcal{G}$, i.e., those groups that contain $x_t$ ($G \ni x_t$), and then proceeds to determine the new treatment probability by aggregating the "best-guess" probabilities for all active groups $G \in \mathcal{G}_t$ determined based on the past performance of those groups. To do so, MGATE maintains group weights $w'_{t,G}$ and group-specific propensities $p_{t,G}$. It comes up with a single effective treatment probability: $p_{t,\text{eff}} \sim \sum_{G \in \mathcal{G}_t} w'_{t,G} p_{t,G}$ in each round by reweighing the group specific propensities of the active groups. This effective treatment probability should simultaneously satisfy the interests of all active groups. The treatment decision $Z_t$ is then generated according to $p_{t,\text{eff}}$. After the outcome is revealed, MGATE updates all group weights, as well as the propensities of groups that were active.

We can show that MGATE achieves the following multi-group Neyman regret guarantee. We note that MGATE is anytime valid, meaning that just like our noncontextual design ClipOGD$^{\text{SC}}$, it does not require advance knowledge of the time horizon $T$.

**Theorem 4.2** (Guarantees for Algorithm 2). *Fix any context space $\mathcal{X}$ and finite group family $\mathcal{G} \subseteq 2^{\mathcal{X}}$. Suppose[4] Assumption 3.1 holds with lower bound constant $c > 0$. Then, for any clipping function $h$, the expected multigroup regret of Algorithm 2 will be bounded as:*

$$\text{RegVarMG}_T(\mathcal{A}; \mathcal{G}) = O\left(\sqrt{|\mathcal{G}|} \cdot (h(T))^5 \cdot \sqrt{T}\right).$$

### 4.3. Technical Overview

The full analysis of Algorithm 2 is contained in Appendix D. It builds on several tools recently developed in the online learning literature, which are formally introduced in Appendix D.1, and we briefly survey them here. The central tool is the sleeping experts algorithmic framework (Blum & Lykouris, 2020), which has recently been shown to be able to combine the wisdom of multiple sub-learners (or experts) into a meta-algorithm with performance on par with each of the sub-learners. The key difference from typical online aggregation schemes is that each sub-learner is allowed to be inactive (asleep) on some rounds, on which it does not

---

[4]By replacing the ClipOGD$^{\text{SC}}$ propensity updates in MGATE with ClipOGD$^0$-style updates, we can straightforwardly obtain a multigroup design which only relies on the assumptions of Dai et al. (2023) while keeping $\widetilde{O}(\sqrt{T})$ multigroup Neyman regret. This follows from the generality of our multigroup meta-design presented in Appendix D, which can use a wide variety "ClipOGD-style" updates while still obtaining $\widetilde{O}(\sqrt{T})$ multigroup regret.

give advice to the meta-algorithm. At a high level, to obtain multigroup Neyman regret, we would thus like to use a sleeping experts algorithm to aggregate propensities suggested by $|\mathcal{G}| = d$ copies of ClipOGD$^{\text{SC}}$ that are respectively active on all groups $G \in \mathcal{G}$; the aggregated design would then perform comparably to each copy of ClipOGD$^{\text{SC}}$ on its group $G$. Then, since that copy of ClipOGD$^{\text{SC}}$ will have no regret on group $G$, neither will the aggregated design.

**Challenges and Solutions** Past work on sleeping experts does not fully address the combination of difficulties present in our setting: (1) stochastic (realized outcome) feedback rather than full-information (both outcomes) feedback; (2) the need to perform clipping of the iterates (propensities) to explicitly restrict them from approaching the feasible set's boundary too fast; and (3) the fact that the gradient feedback magnitude grows unboundedly as $T \to \infty$, even with clipping.

While there are a limited number of "sleeping bandits" algorithms in the literature (e.g., see Nguyen & Mehta (2024)) that address the stochastic feedback, they don't naturally extend to cover both of the latter two issues. Therefore, we design from scratch a new sleeping experts algorithm tailored to all of these challenges. It employs *scale-free* updates of the group weights $w'_t$ so as to control the loss and gradient feedback magnitudes; we achieve this by deploying an instance of the seminal scale-free SOLO FTRL algorithm of Orabona & Pál (2018) and endowing it with sleeping experts regret guarantees via a recent reduction of Orabona (2024). To clip the effective probability magnitudes, our algorithm aggregates over the suggested per-group probabilities via convex combinations rather than via sampling from their mixture. Finally, to ensure that the per-group propensity updates remain valid under stochastic gradient feedback and despite the aggregator using a different propensity than the suggested per-group one, MGATE uses a combination of unbiased first-order ($\widetilde{g}_{t,G}$) and zeroth-order ($\widetilde{\ell}_{t,G}$) per-group feedback estimators, which depend on both $p_{t,\text{eff}}$ and $p_{t,G}$.

**A Generalized Meta-Design** Our analysis in Appendix D generalizes beyond MGATE (Algorithm 2). Indeed, our approach more generally allows the use of any scale-free sleeping experts algorithm to update group weights, and any ClipOGD-style (see Appendix D.3) no-regret adaptive designs to update the groupwise treatment probabilities. Thus, we more generally provide a meta-design that reduces multigroup designs to a broad class of non-contextual, no-regret designs. This generalized meta-design is given as Algorithm 7 in Appendix D.4, and Theorem D.6 contains its regret bound, of which Theorem 4.2 above is a corollary.

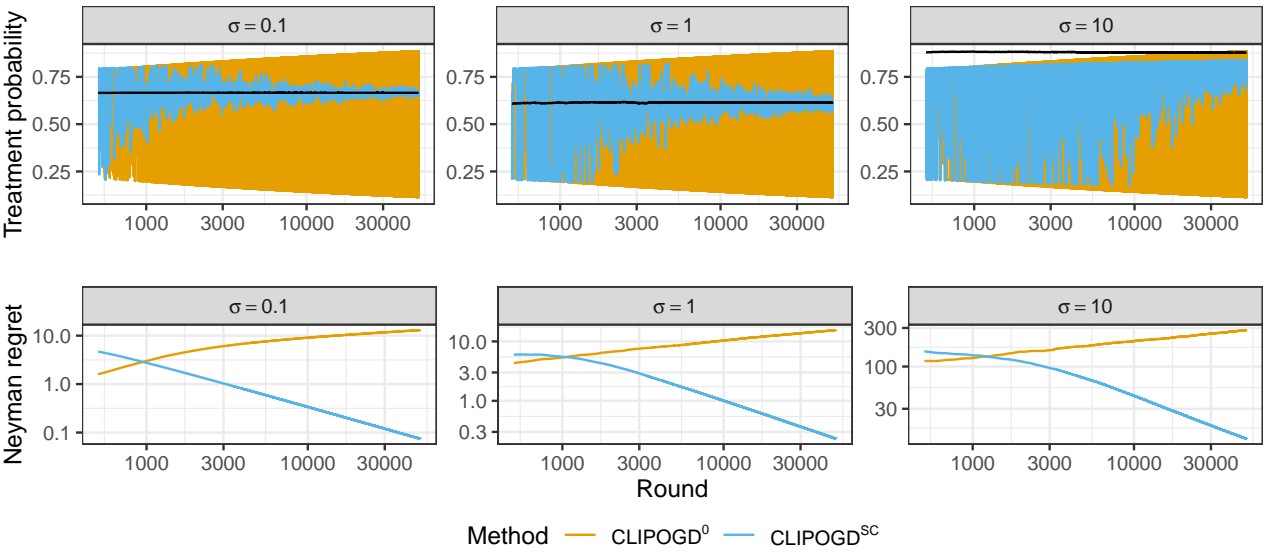

*Figure 1.* **Treatment probabilities and Neyman regret of ClipOGD on Gaussian data** for different noise ($\sigma$) levels. As $\sigma$ increases, ClipOGD$^{\text{SC}}$ converges more slowly. Its regret remains high, and the treatment probabilities do not settle within the observed time horizon ($T \approx 50{,}000$). The black line in the treatment probabilities indicates the Neyman optimal probability.

## 5. Experimental Results

We first present the results for the non-contextual setting and then turn to the analysis of the performance for the contextual algorithm. Our code is available at the following link: https://github.com/amazon-science/adaptive-abtester.

### 5.1. Non-Contextual Experiments

**Tasks** We compare our method ClipOGD$^{\text{SC}}$ with ClipOGD$^0$ (Dai et al., 2023) on multiple tasks. Below, we show two key datasets (one synthetic and one real-world) used in our experiments, with full details in Appendix E. The first is a synthetic dataset is generated as follows: $y_t(i) \overset{\text{iid}}{\sim} \mathcal{N}(\mu_i, \sigma^2)$ for $t = 1, \ldots, T$ and $i = 0, 1$ with $\mu_0 = 1$ and $\mu_1 = 2$. We vary $\sigma_i \in \mathbb{R}_+$ to showcase where our method succeeds and where it struggles. The second dataset comes from Egypt's largest microfinance organization (Groh & McKenzie, 2016), covering 2,961 clients. Here, the treatment is a new insurance product, and the outcome is how much individuals invest in machinery. Following Dai et al. (2023), we fill missing values with Gaussian noise and resample each unit five times to increase the population size. We also present experiments on the ASOS Digital Experiments Dataset (Liu et al., 2021), and on question-answering tasks for large language models, including BigBench (Srivastava et al., 2023), in the Appendix.

**Experimental Setup** In our simulation, each unit is randomly assigned to treatment or control using the treatment probability from our method or ClipOGD$^0$. We repeat this process 10,000 times, generating many different treatment-control paths. We then measure the Neyman regret by averaging the regret across these probabilities obtained at each time step.

**Hyperparameter Choices** Throughout the experiments, we use the following hyperparameters. For our method, we set $\eta_t = 2/t$ and $\delta_t = 1/h(t)$, where the clipping function is $h(t) = \exp\big((\log(t+2))^{1/4}\big)$. For ClipOGD$^0$, we follow Dai et al. (2023) with a constant learning rate $\eta_t = 1/\sqrt{T}$ and clipping rate $\delta_t = 0.5 \cdot t^{-1/\sqrt{5 \log T}}$.

**Results** We analyze three synthetic data settings where we vary $\sigma$ as $\{0.1, 1, 10\}$. As $\sigma$ increases, the ratio $C/c$ also grows, so by Equation (1), we expect slower convergence of our algorithm. We set $T = 50{,}000$. Figure 1 shows the Neyman regret across these settings, matching our theoretical expectations: when $\sigma = 0.1$, the regret of ClipOGD$^{\text{SC}}$ drops to 0 quickly, but for larger $\sigma$, the regret remains high and converges later. The regret of ClipOGD$^0$ instead keeps increasing with time. Nonetheless, in line with Corollary 3.4, Figure 1 also shows that our method's adaptively chosen propensities ultimately converge to the Neyman optimal probability in all three cases. By contrast, the propensities of ClipOGD$^0$ only converge when $\sigma = 10$, which happens to match the initial probability of 0.5. Next, we turn to examine the results on the microfinance data. Figure 2 illustrates the treatment probabilities and Neyman regret for both algorithms. On average, each design assigns probabilities near the Neyman probability. However, those of ClipOGD$^0$ exhibit higher variance compared to ClipOGD$^{\text{SC}}$. This translates into greater Neyman regret in

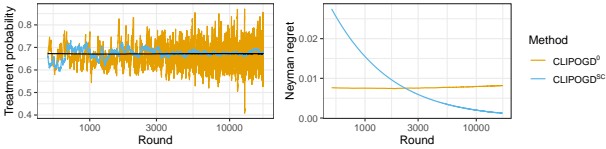

*Figure 2.* **Treatment probabilities and Neyman regret of ClipOGD on microfinance data** for $T \approx 15{,}000$ rounds.

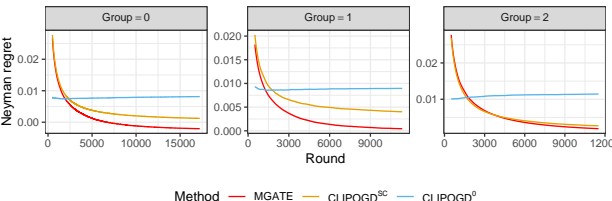

*Figure 3.* **Group-conditional Neyman regret of ClipOGD and MGATE on microfinance data**. MGATE produces the lowest $\mathcal{G}$-*multigroup* Neyman regret as desired, and in this case dominates the non-contextual ClipOGD variants for each group, including the noncontextual group $G_0 = \mathcal{X}$.

later rounds, which never converges to 0. The probabilities assigned by our method, instead, converge to the Neyman probability, yielding vanishing average Neyman regret.

### 5.2. Contextual Experiments

Here we present our contextual results using Algorithm 2 over the previously-described datasets. To standardize the contextual groups in each experiment, we design simple, synthetic post-hoc groups by scoring each sample as $s_t = 1 / \left(1 + \frac{y_t(0)^2}{y_t(1)^2 + \epsilon}\right)$ (the optimal Neyman sampling probability for the single sample). Our groups are computed by checking whether sample $t$ belongs to some predetermined quantile of the score function $G_0 = \mathcal{X}, G_1 = \mathbb{1}\left[F^{-1}(s_t) \leq \frac{2}{3}\right], G_2 = \mathbb{1}\left[\frac{1}{3} \leq F^{-1}(s_t)\right]$. We note that these groups are overlapping and informative since $G_1$ is guaranteed to have lower or equal optimal sampling probability than $G_2$.

We stress that these groups are included for illustrative purposes and rely on information that would be unobservable in a real ATE experiment, but nonetheless showcase the potential for high-quality contextual information for multi-group ATE. Figure 3 shows the Neyman regret for ClipOGD[0], ClipOGD[SC], and MGATE on the microfinance dataset on each group; our MGATE method achieves the lowest group-conditional regret out of all the methods, effectively minimizing the $\mathcal{G}$-*multigroup* Neyman regret, and thereby validating our theoretical results. Additional contextual experiments are provided in the Appendix.

## 6. Conclusion

In this paper, we have studied adaptive designs for unbiased ATE estimation with finite-population guarantees. We introduced a modification of the ClipOGD algorithm that provably yields vanishing Neyman regret, achieving an anytime-valid $\widetilde{O}(\log T)$ Neyman regret, improving upon previous $\widetilde{O}(\sqrt{T})$ guarantees. We also extend our framework to incorporate contextual information by introducing a multigroup formulation. Our proposed multigroup adaptive design ensures $\widetilde{O}(\sqrt{T})$ regret for each predefined group, enabling efficiency improvements for subgroup ATE estimation. Experimental results corroborate these findings.

Overall, these results suggest that adaptive experimentation can achieve strong finite-population efficiency guarantees, offering practical advantages for a wide range of applications. Future work could explore extensions to other experimental designs and further reductions in regret rates.

## Acknowledgements

The authors thank Vanessa Murdock for the support throughout this project, and Lorenzo Masoero, Blake Mason, and James McQueen for useful feedback.

## Impact Statement

This paper presents work whose goal is to advance the field of Machine Learning. There are many potential societal consequences of our work, none which we feel must be specifically highlighted here.

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

## Organization

The Appendix is organized as follows.

- Appendix A contains a discussion of further related work.

- Appendix B contains proofs of our noncontextual method's convergence.

- Appendix C discusses confidence interval guarantees for adaptive IPW estimators induced by our design.

- Appendix D presents the general multigroup adaptive design framework and proves its efficiency guarantees.

- Appendix E describes additional empirical results.

## A. Additional Related Work

**Some early work on adaptive designs**    The efficient adaptive design problem considered in this work (and in (Dai et al., 2023) and other prior works) has its roots in problems studied, or mentioned, in the classical works of Neyman, Wald, and Robbins. The seminal work of Wald (1992) is the historical underpinning of much of the research in sequential experimental design, particularly on the hypothesis testing side. Our work shares the broad motivation of increasing statistical efficiency through adaptivity, though we do not explicitly deal with some of the notions that Wald focused on such as early stopping.

As described in (Dai et al., 2023), it was Robbins (1952) who explicitly posed the open problem of constructing efficient adaptive sampling schemes, though he stopped short of formally reasoning about precise benchmarks such as Neyman regret that such designs must optimize; and while within the subsequent decades, adaptive designs have gained traction e.g. in the survey sampling context Solomon & Zacks (1970), but yet again they did not explicitly optimize for efficiency metrics like the ones we study.

**Improved Neyman Regret in the Superpopulation Setting**    Independently and concurrently to our work, complementary progress has been made on the problem of (noncontextual) adaptive ATE estimation with low Neyman regret. Neopane et al. (2024) modify ClipOGD to obtain logarithmic Neyman regret guarantees in the substantially more benign superpopulation regime. Neopane et al. (2025), in an environment in which outcomes are generated from some joint distribution with time-stationary means and variances, design a Neyman regret minimization algorithm based on a UCB-like optimistic policy tracking approach, with both theoretical guarantees and strong empirical performance.

**Covariate Balancing**    In relation to our multigroup setting, it is interesting to discuss recent progress on what is known as *covariate balancing* techniques. For instance, studies such as Harshaw et al. (2024), Rao & Zhang (2024), which are conducted in the non-adaptive setting, share with our work the objective of optimizing the estimation variance — and offer a principled way to exploit covariates to do so, by assuming the mapping between covariates and outcomes is linear (and trading off robustness and covariate balance). Thus, a potential future work direction can involve porting the covariate balancing insights over to the adaptive setting, possibly leading to a new perspective on group-aware optimal-variance adaptive designs that could complement our multigroup approach.

## B. Non-Contextual Setting: Proof of Theorem 3.2 and of Lemma 3.3

### B.1. Neyman Regret Analysis for ClipOGD$^{\text{SC}}$ : Proof of Theorem 3.2

We establish Theorem 3.2 via a sequence of claims.

*Claim* 1 (Optimal Probability Bounds; Lemma C.2 of Dai et al. (2023)).  The optimal fixed probability $p_T^*$ for any time horizon $T$ satisfies, under Assumption 3.1, the following inequality, defining the constant $A = 1 + C/c \geq 2$:

$$\frac{1}{A} \leq p_T^* \leq 1 - \frac{1}{A}.$$

*Claim* 2 (How Quickly Optimal Probability Enters Admissible Region).  Under Assumption 3.1, let $A = 1 + C/c \geq 2$. Then, for any time horizon $T$, the optimal probability $p_T^*$ will satisfy:

$$t \geq t^* \implies p_T^* \in [\delta_t, 1 - \delta_t], \quad \text{where } t^* := h_{\text{inv}}(A).$$

*Proof.* With 1 in hand, we have that as soon as $\delta_t \leq 1/A$, the optimal probability $p_T^*$ (for any $T$) is guaranteed to be in the admissible interval $[\delta_t, 1 - \delta_t]$. This is equivalent to requiring $h(t) \geq A$, which by definition of $h_{\text{inv}}$ and by the strictly increasing nature of $h$ is equivalent to $t \geq h_{\text{inv}}(A)$. $\qquad\square$

*Claim* 3 (Gradient Raw Moment Bounds). Under Assumption 3.1, for every $t \geq 1$ we have the following bounds in expectation wrt. the design's randomness:

$$\mathbb{E}[|g_t|] \leq 2C^2 h(t)^2, \quad \mathbb{E}[g_t^2] \leq 2C^4 h(t)^5.$$

*Proof.* The bounds follow as shown in Lemma C.5 of Dai et al. (2023), by just expanding out the first and second raw absolute moment of the gradient estimator defined above; we will get $\mathbb{E}[|g_t|] \sim \delta_t^{-2}(y_t(1)^2 + y_t(0)^2)$, and $\mathbb{E}[g_t^2] \sim \delta_t^{-5}(y_t(1)^4 + y_t(0)^4)$, so the statement follows from our Assumption 3.1, or from Dai et al. (2023)'s assumption on the boundedness of the second and fourth moments of the two populations. $\qquad\square$

*Claim* 4 (Strong Convexity of Objective). For any round $t \geq 1$, and for any $p, p' \in (0, 1)$, the objective function will satisfy:

$$f_t(p) - f_t(p') \leq f'(p) \cdot (p - p') - c^2(p - p')^2.$$

*Proof.* To show this, it suffices to establish $2c^2$-strong convexity of $f_t(p) = \frac{y_t(0)^2}{p} + \frac{y_t(1)^2}{1-p}$, and we will do so by verifying that $f''(p) \geq 2c^2$ for all $p \in (0, 1)$. Indeed, note that $f''(p) = 2\left(\frac{y_t(0)^2}{p^3} + \frac{y_t(1)^2}{(1-p)^3}\right) \geq 2(y_t(0)^2 + y_t(1)^2) \geq 2c^2$ since $p \in (0, 1)$ and by definition of $c$ in Assumption 3.1. $\qquad\square$

*Claim* 5. For any $t \geq 1$, any setting of $\eta_t > 0$, $\delta_t = 1/h(t)$, and for any point $p^* \in \{p_t^*\}_{t \geq 1}$, we have in expectation over the randomness of the design:

$$\mathbb{E}[f_t(p_t) - f_t(p^*)] \leq \left(\frac{1}{2\eta_t} - c^2\right) \mathbb{E}[(p_t - p^*)^2] - \frac{1}{2\eta_t} \mathbb{E}[(p_{t+1} - p^*)^2] + \eta_t \cdot (Ch(t))^5$$

$$+ 2 \cdot 1[t \leq t^*] \cdot \left(\frac{1}{\eta_t \cdot h(t)} + (Ch(t))^2\right).$$

*Proof.* By Claim 4 applied to $p = p_t$ and $p' = p^*$, we have $f_t(p_t) - f_t(p^*) \leq f'(p_t) \cdot (p_t - p^*) - c^2(p_t - p^*)^2$. Now, we can bound the first term on the right-hand side as follows.

First, start with the inequality: $|p_{t+1} - p^*| \leq |p_t - \eta_t g_t - p^*| + \delta_t \cdot 1[p^* \notin [\delta_t, 1 - \delta_t]]$, which follows by Lemma C.1 in Dai et al. (2023). By Claim 2, we have that $1[p^* \notin [\delta_t, 1 - \delta_t]] = 0$ for all $t \geq t^*$, implying that $1[p^* \notin [\delta_t, 1 - \delta_t]] \leq 1[t \leq t^*]$. Thus, we have $|p_{t+1} - p^*| \leq |p_t - \eta_t g_t - p^*| + \delta_t \cdot 1[t \leq t^*]$. Squaring this inequality, we arrive, after rearranging terms and using the triangle inequality, at

$$(p_{t+1} - p^*)^2 \leq (p_t - p^*)^2 + \eta_t^2 g_t^2 - 2\eta_t g_t(p_t - p^*) + 4 \cdot 1[t \leq t^*] \cdot \eta_t \cdot \delta_t \left(\frac{1}{\eta_t} + \frac{|g_t|}{2}\right).$$

Rearranging terms once again, we get:

$$2\eta_t g_t(p_t - p^*) \leq (p_t - p^*)^2 + \eta_t^2 g_t^2 - (p_{t+1} - p^*)^2 + 4 \cdot 1[t \leq t^*] \cdot \eta_t \cdot \delta_t \left(\frac{1}{\eta_t} + \frac{|g_t|}{2}\right).$$

Dividing this by $\eta_t > 0$, we get:

$$2g_t(p_t - p^*) \leq \frac{1}{\eta_t}\left((p_t - p^*)^2 - (p_{t+1} - p^*)^2\right) + \eta_t g_t^2 + 4 \cdot 1[t \leq t^*] \cdot \delta_t \left(\frac{1}{\eta_t} + \frac{|g_t|}{2}\right).$$

Noting that $\mathbb{E}[g_t|\mathcal{F}_t] = f_t'(p_t)$ by definition of $g_t$, as well as using the bounds on the expected gradient moments from Claim 3, we can take the expectation of the last inequality to obtain:

$$2f_t'(p_t)(p_t - p^*) \leq \frac{1}{\eta_t}\left((p_t - p^*)^2 - \mathbb{E}[(p_{t+1} - p^*)^2|\mathcal{F}_t]\right) + \eta_t \mathbb{E}[g_t^2|\mathcal{F}_t] + 4 \cdot 1[t \leq t^*] \cdot \delta_t \left(\frac{1}{\eta_t} + \frac{\mathbb{E}[|g_t||\mathcal{F}_t]}{2}\right)$$

$$\leq \frac{1}{\eta_t}\left((p_t - p^*)^2 - \mathbb{E}[(p_{t+1} - p^*)^2|\mathcal{F}_t]\right) + \eta_t \cdot 2C^4 h(t)^5 + 4 \cdot 1[t \leq t^*] \cdot \delta_t \left(\frac{1}{\eta_t} + C^2 h(t)^2\right).$$

Returning to the strong convexity-induced inequality above, we thus have:

$$
\begin{aligned}
f_t(p_t) - f_t(p^*) &\leq f'(p_t) \cdot (p_t - p^*) - c^2(p_t - p^*)^2 \\
&\leq \frac{1}{2\eta_t}\left((p_t - p^*)^2 - \mathbb{E}[(p_{t+1} - p^*)^2|\mathcal{F}_t]\right) + \eta_t \cdot C^4 h(t)^5 \\
&\quad + 2 \cdot 1[t \leq t^*] \cdot \delta_t \left(\frac{1}{\eta_t} + C^2 h(t)^2\right) - c^2(p_t - p^*)^2 \\
&= \left(\frac{1}{2\eta_t} - c^2\right)(p_t - p^*)^2 - \frac{1}{2\eta_t}\mathbb{E}[(p_{t+1} - p^*)^2|\mathcal{F}_t] + \eta_t \cdot C^4 h(t)^5 \\
&\quad + 2 \cdot 1[t \leq t^*] \cdot \delta_t \cdot \left(\frac{1}{\eta_t} + C^2 h(t)^2\right).
\end{aligned}
$$

Now, taking expectation again, now with respect to the randomness up through $\mathcal{F}_t$, we obtain the statement of this claim. $\square$

*Claim* 6 (Convergence Bound). For any time horizon $T$, and any $p^* \in \{p_t^*\}_{t \geq 1}$, we have:

$$\sum_{t=1}^{T} \mathbb{E}[f_t(p_t) - f_t(p^*)]$$

$$\leq -c^2(T+1)\,\mathbb{E}[(p_{T+1} - p^*)^2] + \frac{C^5}{2c^2}h(T)^5(\log(T+1)+1) + 2C^2\left(1+\frac{C}{c}\right)^2 h_{\text{inv}}\left(1+\frac{C}{c}\right)$$

$$+ 2c^2\left(h_{\text{inv}}\left(1+\frac{C}{c}\right)+1\right)^2.$$

*Proof.* Summing the inequality in Claim 5 from $t = 1$ to $t = T$, we obtain via telescoping sums:

$$\sum_{t=1}^{T} \mathbb{E}[f_t(p_t) - f_t(p^*)]$$

$$
\begin{aligned}
&\leq \sum_{t=1}^{T}\left(\frac{1}{2\eta_t} - c^2\right)\mathbb{E}[(p_t - p^*)^2] - \sum_{t=1}^{T}\frac{1}{2\eta_t}\mathbb{E}[(p_{t+1} - p^*)^2] + \sum_{t=1}^{T}\eta_t \cdot (Ch(t))^5 \\
&\quad + \sum_{t=1}^{T} 2 \cdot 1[t \leq t^*] \cdot \left(\frac{1}{\eta_t \cdot h(t)} + (Ch(t))^2\right) \\
&\leq \sum_{t=1}^{T}\left(\frac{1}{2\eta_t} - c^2\right)\mathbb{E}[(p_t - p^*)^2] - \sum_{t=1}^{T}\frac{1}{2\eta_t}\mathbb{E}[(p_{t+1} - p^*)^2] + \sum_{t=1}^{T}\eta_t \cdot (Ch(t))^5 \\
&\quad + 2\sum_{t=1}^{t^*}\left(\frac{1}{\eta_t \cdot h(t)} + (Ch(t))^2\right) \\
&\leq \sum_{t=1}^{T}\left(\frac{1}{2\eta_t} - c^2\right)\mathbb{E}[(p_t - p^*)^2] - \sum_{t=1}^{T}\frac{1}{2\eta_t}\mathbb{E}[(p_{t+1} - p^*)^2] \\
&\quad + (Ch(T))^5\sum_{t=1}^{T}\eta_t + 2t^* \cdot (Ch(t^*))^2 + 2\sum_{t=1}^{t^*}\frac{1}{\eta_t \cdot h(t)} \\
&= \left(\frac{1}{2\eta_1} - c^2\right)\mathbb{E}[(p_1 - p^*)^2] - \frac{1}{2\eta_{T+1}}\mathbb{E}[(p_{T+1} - p^*)^2] \\
&\quad + \sum_{t=2}^{T}\left(\frac{1}{2\eta_t} - \frac{1}{2\eta_{t-1}} - c^2\right)\mathbb{E}[(p_t - p^*)^2] + (Ch(T))^5\sum_{t=1}^{T}\eta_t + 2t^* \cdot (Ch(t^*))^2 + 2\sum_{t=1}^{t^*}\frac{1}{\eta_t \cdot h(t)}
\end{aligned}
$$

$$\leq -c^2(T+1)\,\mathbb{E}[(p_{T+1}-p^*)^2] + \frac{(Ch(T))^5}{2c^2}(\log(T+1)+1) + 2t^* \cdot (Ch(t^*))^2 + 4c^2 \sum_{t=1}^{t^*} \frac{t}{h(t)}.$$

Finally, recalling the definition of $t^* = h_{\text{inv}}(A) = h_{\text{inv}}(1+C/c)$ and substituting it in, we obtain the desired claim. $\qquad\square$

Finally, with the result of Claim 6 in hand, we observe that (1) the term $-c^2(T+1)\,\mathbb{E}[(p_{T+1}-p^*)^2]$ is nonpositive and can thus be ignored, (2) the second term on the right hand side is asymptotically $O((h(T))^2 \cdot \log T)$, and (3) the third and fourth terms on the right hand side are constant with respect to $T$ and only a function of the constants $C, c$ of the problem. This gives the desired result.

## B.2. Convergence of Treatment Probabilities of ClipOGD$^{\text{SC}}$ : Proof of Lemma 3.3

We will make use of Claim 6 from the previous subsection. Simply rearranging the terms, we obtain the following bound for the deterministic setting:

$$c^2(T+1)\,\mathbb{E}[(p_{T+1}-p_T^*)^2] \leq -\sum_{t=1}^{T} \mathbb{E}[f_t(p_t) - f_t(p_T^*)] + \frac{C^2}{2c^2}h(T)^2(\log(T+1)+1) + O(1),$$

where the $O(1)$ term hides terms in the bound that do not depend on $T$. Dividing through by $c^2 \cdot (T+1)$ and reindexing for convenience, we obtain the desired result:

$$\mathbb{E}[(p_T - p_T^*)^2] \leq -\Theta\left(\frac{\mathbb{E}[\text{Reg}_T]}{T}\right) + O\left(\frac{(h(T))^2 \log T}{T}\right).$$

## C. Confidence Interval Guarantees: Proof Sketch for Theorem 3.7

*Remark* C.1 (Chebyshev vs. Wald Confidence Intervals). As Dai et al. (2023) point out, it appears that ClipOGD may lead to an asymptotically normal distribution of the IPW estimator. If this were true, that would allow us to get Wald-type confidence intervals for the IPW estimator based on the variance estimator $\widehat{\text{VB}}$, which would be narrower than Chebyshev-type ones. Through some simulations, we observed that asymptotically, the z-score of the IPW estimator induced by our adaptive scheme appears to satisfy asymptotic normality. However, below we only prove the validity of Chebyshev-type confidence intervals, and leave Wald-type CIs to be explored in future work. $\qquad\square$

We will convince ourselves that the techniques employed in Dai et al. (2023) for proving the validity of this variance estimator apply to a broad class of adaptive sampling schemes. Dai et al. (2023) state this result for their particular adaptive design but mention that it may apply to other learning rate and clipping rate settings. And indeed, we find that while their approach does depend on the adaptive design having sufficiently slowly decaying clipping rate and vanishing Neyman regret, it is oblivious to hyperparameters such as the learning rate. Moreover, we find that the condition of having asymptotically nonnegative Neyman regret, which Dai et al. (2023) impose on the design, is also not necessary to ensure that the variance estimator $\widehat{\text{VB}}$ is conservatively valid.

For easier tracking of the relevant quantities, recall the notation: $S_T(i) := \sqrt{\frac{1}{T}\sum_{t=1}^{T} y_t(i)^2}$ for $i \in \{0,1\}$. Following (Dai et al., 2023), we define the quantities $A_T(1) = (S_T(1))^2$, $A_T(0) = (S_T(0))^2$, as well as the quantities $\widehat{A_T(1)} = \frac{1}{T}\sum_t y_t(1)^2 \frac{Z_t}{p_t}$, $\widehat{A_T(0)} = \frac{1}{T}\sum_t y_t(0)^2 \frac{1-Z_t}{1-p_t}$ that estimate them in an unbiased way. Recalling that the variance of the optimal nonadaptive design (i.e., the variance of the IPW estimator that uses $p_T^*$ as its fixed sampling probability on all rounds $t = 1 \ldots T$) is

$$\frac{2}{T}(1+\rho)S_T(1)S_T(0) \leq \text{VB} := \frac{4}{T}\sqrt{A_T(1)A_T(0)},$$

we can see that $\widehat{\text{VB}} = \frac{4}{T}\sqrt{\widehat{A_T(1)}\widehat{A_T(0)}}$ simply aims to approximate the upper bound VB on the optimal fixed-probability sampling scheme's variance. And given that our design has a no-regret guarantee with respect to this benchmark, $\widehat{\text{VB}}$ thus also asymptotically approximates the upper bound on our (and any other such) design's induced IPW estimator variance $V_T$. This is the blueprint of the proof, and we will now briefly revisit the technical steps in Dai et al. (2023) that make this blueprint argument work.

First, Proposition D.1 of Dai et al. (2023) proves that

$$\left| \mathbb{E}[\widehat{A_T(1)}\widehat{A_T(0)}] - A_T(1)A_T(0) \right| \leq \frac{C^4}{T},$$

which by tracking the proof can be seen to not depend on the sampling scheme.

Second, by generalizing the result and steps of Proposition D.2 of Dai et al. (2023), we can bound the variance of the (normalized version of the) estimator $\widehat{\mathrm{VB}}$ as:

$$\mathrm{Var}(\widehat{A_T(1)}\widehat{A_T(0)}) \leq \frac{C^8 \cdot h(T)}{T} + \frac{C^8 \cdot (h(T))^2}{T^2} \leq \frac{2C^8 \cdot h(T)}{T}.$$

Thus, applying Chebyshev's inequality to this variance bound and using the preceding in-expectation bound, we conclude that $\widehat{A_T(1)}\widehat{A_T(0)} \to A_T(1)A_T(0)$ in probability at the rate $O_p((h(T)/T)^{1/2})$.

Now, as in the proof of Theorem 5.1 of Dai et al. (2023), we can observe that a Continuous Mapping Theorem can be applied to this in-probability convergence result to give the implication that $\sqrt{\widehat{A_T(1)}\widehat{A_T(0)}} \to \sqrt{A_T(1)A_T(0)}$ at the same asymptotic rate $O_p((h(T)/T)^{1/2})$. Indeed, since the target random variable $A_T(1)A_T(0)$ is bounded below by $c^2$ by Assumption 3.1, the square root transformation will be Lipschitz on the relevant range (i.e., away from zero).

Finally, to establish the validity of the Chebyshev-type confidence intervals given above, it suffices to look at the z-score statistic $\zeta = \frac{\tau_T - \hat{\tau}_T}{\sqrt{\mathrm{Var}(\hat{\tau}_T)}}$ and the estimated z-score statistic $\zeta' = \frac{\tau_T - \hat{\tau}_T}{\sqrt{\widehat{\mathrm{VB}}}}$ and establish that $\zeta$ stochastically dominates $\zeta'$. Towards this, note as in Dai et al. (2023) that:

$$\zeta' = \zeta \cdot \left( \sqrt{\frac{\mathrm{Var}(\hat{\tau}_T)}{\mathrm{VB}}} \cdot \sqrt{\frac{T \cdot \mathrm{VB}}{T \cdot \widehat{\mathrm{VB}}}} \right).$$

First, since the estimator $\hat{\tau}_T$ is induced by a no-regret adaptive design and since VB is an upper bound on the variance of the best fixed SRS scheme (which serves as the benchmark of the design's regret performance), we have that $\limsup_{T \to \infty} \frac{\mathrm{Var}(\hat{\tau}_T)}{\mathrm{VB}} \leq 1$. Second, from what we just obtained, $T \cdot \widehat{\mathrm{VB}} \to T \cdot \mathrm{VB}$ in probability, which in view of $T \cdot \widehat{\mathrm{VB}}$ being lower-bounded by a constant by Assumption 3.1 implies by the Continuous Mapping Theorem that $\sqrt{\frac{T \cdot \mathrm{VB}}{T \cdot \widehat{\mathrm{VB}}}}$ converges to 1 in probability. By Slutsky's theorem, this proves the desired stochastic domination and thus implies that the proposed confidence interval construction is asymptotically (conservatively) valid.

# D. Multigroup Adaptive Design: Proofs and Details

## D.1. OLO Primitives

Our multigroup design will rely on a sequence of reductions, derived with the help of some online learning machinery: a recent reduction of Orabona (2024) and scale-free algorithms by Orabona & Pál (2018). First, we spell out the algorithmic primitives that we will require.

**Definition D.1** (OLO algorithm; OLO regret). An *OLO (online linear optimization) algorithm* $\mathcal{A}$ over domain $V \subseteq \mathbb{R}^d$, where $d \geq 1$ is the dimension of the problem, sequentially receives vectors $\ell_t \in \mathbb{R}^d$, $t = 1, 2, \ldots$. Each $\ell_t$ is interpreted as the "gradient", or the "loss", that $\mathcal{A}$ suffers at round $t$.

Each round, before seeing $\ell_t$, algorithm $\mathcal{A}$ outputs iterate $v_t \in V$ as a function of past history. The algorithm's *regret* at any time $T$ is defined as the total loss incurred by its iterates minus the total loss of the best-in-hindsight admissible solution:

$$\mathrm{Reg}_T(\mathcal{A}) := \max_{v \in V} \mathrm{Reg}_T(\mathcal{A}; v), \quad \text{where } \mathrm{Reg}_T(\mathcal{A}; v) = \sum_{t=1}^{T} \langle \ell_t, v_t - v \rangle \text{ for } v \in V.$$

**Definition D.2** (Sleeping Experts algorithm; SE regret). A *sleeping experts (SE) algorithm* $\mathcal{A}$ over domain $V \subseteq \mathbb{R}^d$, where $d \geq 1$ is the number of "sleeping experts", sequentially receives vectors $a_t \in \{0,1\}^d$ and $\ell_t \in \mathbb{R}^d$ at rounds $t = 1, 2, \ldots$. The vector $a_t$ has the interpretation that $a_{t,i} \in \{0,1\}$ (for each $i \in [d]$) denotes whether expert $i$ is "active" (1) or "inactive"

(0) in round $t$. The vector $\ell_t$ has the interpretation that at any round $t$, for all active experts $i$ (i.e., $a_{t,i} = 1$), expert $i$'s loss is $\ell_{t,i}$, while for all inactive experts $i$ the loss coordinate $\ell_{t,i}$ is (arbitrarily) equal to 0.

Each round, after seeing $a_t$ but before seeing $\ell_t$ (i.e., after seeing which experts are active but before seeing their losses), the algorithm outputs a distribution $v_t \in \Delta_d$ as a function of past history, such that $v_{t,i} = 0$ for all inactive experts (i.e., for all $i$ such that $a_{t,i} = 0$). In words, at each round the algorithm is required to output a distribution $v_t$ over the currently active experts only.

The algorithm's *Sleeping Experts regret* at any time $T$ is defined as the upper bound, over all experts $i \in [d]$, on its performance relative to expert $i$ over those rounds $t$ *on which $i$ was active*:

$$\text{RegSE}_T(\mathcal{A}) := \max_{i \in [d]} \sum_{t=1}^{T} a_{t,i} \cdot (\langle \ell_t, v_t \rangle - \ell_{t,i}).$$

**Scale-Free OLO** We will make use of a *scale-free* OLO algorithm (Orabona & Pál, 2018) to design a base algorithm for our multigroup regret algorithm. The property of any such algorithm is that its regret bound does not require the norms of the gradients $\ell_t$ to be bounded in $[0, 1]$ for some norm (like standard OLO methods typically require).

*Fact* 1 (Theorem 1 of (Orabona & Pál, 2018)). Fix any norm $\|\cdot\|$ and its dual norm $\|\cdot\|_*$. Then, Algorithm 3 called SOLO FTRL achieves, for any convex closed set $V \subseteq \mathbb{R}^d$, the following regret bound *to any point $v \in V$* that scales with the magnitude of the losses/gradients:

$$\text{Reg}_T(\text{SOLO FTRL}; v) \le (R(v) + 2.75)\sqrt{\sum_{t=1}^{T} \|\ell_t\|_*^2} + 3.5 \min \left\{ \sqrt{T-1}, \text{diam}(V) \right\} \max_{t \in [T]} \|\ell_t\|_*.$$

where $\text{diam}(V) = \sup_{v_1, v_2 \in V} \|v_1 - v_2\|$, and where SOLO FTRL is parameterized by an arbitrary nonnegative continuous 1-strongly-convex regularizer $R : V \to \mathbb{R}$.

---

**Algorithm 3** $\mathcal{A}_{SOLO}$: SOLO FTRL (Orabona & Pál, 2018)

---

Receive domain $V \subseteq \mathbb{R}^d$ base regularizer $R(w)$, and norm $\|\cdot\|$.
Initialize $L_0 \leftarrow \mathbf{0}^d, q_0 \leftarrow 0$.
**for** $t = 1, 2, \dots$ **do**
    Compute new weights $w_t \leftarrow \arg\min_{w \in V} \{\langle L_{t-1}, w \rangle + R_t(w)\}$, where $R_t(w) = \sqrt{q_{t-1}} \cdot R(w)$.
    Receive loss vector $\ell_t$.
    Set $L_t \leftarrow L_{t-1} + \ell_t$.
    Set $q_t \leftarrow q_{t-1} + \|\ell_t\|_*^2$.
**end for**

---

### D.2. Designing a Scale-Free Sleeping Experts Algorithm

Now, let us instantiate the above Fact 1 appropriately. First, set the norm for the regret bound to be the 2-norm: $\|\cdot\| = \|\cdot\|_* = \|\cdot\|_2$. Second, set the regularizer to be $R(v) := \|v\|_2^2$ for $v \in V$, which is 1-convex with respect to the 2-norm. Third, set the domain of the algorithm to be the non-negative orthant: $V = \mathbb{R}_{\ge 0}^d$. We then arrive at the following guarantee.

**Corollary D.3** (of Fact 1). *With the nonnegative orthant $V = \mathbb{R}_{\ge 0}^d$ as domain and the squared $L_2$-norm as regularizer, SOLO FTRL achieves the following scale-free regret bound for all $v \in V$:*

$$\text{Reg}_T(\text{SOLO FTRL}; v) \le \left( \|v\|_2^2 + 6.25 \right) \max_{t \in [T]} \|\ell_t\|_2 \sqrt{T}.$$

*The instantiation of SOLO FTRL for these specific choices is given in Algorithm 4.*

We note that the update for $w_t$ in Algorithm 4 is the solution to the original argmax problem in Algorithm 3, with the nonnegative orthant as domain and the rescaled $L_2$-norm as regularizer.

---

**Algorithm 4** $\mathcal{A}_{SOLO}$: Instantiation for scale-free sleeping experts

---

1: Initialize $L_0 \leftarrow \mathbf{0}^d, q_0 \leftarrow 0$.
2: **for** $t = 1, 2, \ldots$ **do**
3:     Set weights $w_t \leftarrow \max\left\{\mathbf{0}^d, -\frac{1}{\sqrt{q_{t-1}}}L_{t-1}\right\}$ (coordinate-wise maximum).
4:     Receive loss vector $\ell_t \in \mathbb{R}^d_{\geq 0}$.
5:     Set $L_t \leftarrow L_{t-1} + \ell_t$.
6:     Set $q_t \leftarrow q_{t-1} + \|\ell_t\|_2^2$.
7: **end for**

---

**Scale-Free Sleeping Experts**    Now, we will turn this just obtained scale-free OLO regret guarantee into a scale-free sleeping experts regret guarantee. We will utilize a recent black-box reduction mechanism of Orabona (2024), which proceeds as follows.

*Fact* 2 (Sleeping Experts to OLO Reduction (Orabona, 2024)). Consider a sleeping experts setting with $d$ experts. Define any base OLO algorithm $\mathcal{A}$ with nonnegative orthant $V = \mathbb{R}^d_{\geq 0}$ as the domain. Then Algorithm 5, which we refer to as $\mathcal{A}_{OLO \rightarrow SE}$, constructs a sequence $v_1, v_2, \ldots$ of distributions over active experts that attains the following sleeping experts regret bound:

$$\text{RegSE}_T(\mathcal{A}_{OLO \rightarrow SE}) = \max_{v \in \text{SB}(\mathbb{R}^d)} \text{Reg}_T\left(\mathcal{A}\left(\left\{\widetilde{\ell}_t\right\}_{t \in [T]}\right); v\right).$$

Here, $\text{SB}(\mathbb{R}^d)$ as the collection of the $d$ standard basis (unit) vectors of $\mathbb{R}^d$; and the vectors $\{\widetilde{\ell}_t\}_{t \in [T]}$, defined in Algorithm 5, are surrogate loss vectors. Note that these surrogate losses satisfy $\left\|\widetilde{\ell}_t\right\|_\infty \leq 2\|\ell_t\|_\infty$ relative to the original losses $\{\ell_t\}_{t \in [T]}$.

---

**Algorithm 5** $\mathcal{A}_{OLO \rightarrow SE}$: Sleeping Experts to OLO Reduction (Orabona, 2024)

---

    Initialize any base OLO algorithm $\mathcal{A}$ with nonnegative orthant $V = \mathbb{R}^d_{\geq 0}$ as domain.
    **for** $t = 1, 2, \ldots$ **do**
        Get unscaled prediction $w_t \in \mathbb{R}^d_{\geq 0}$ from $\mathcal{A}$.
        Receive indicator vector describing which experts are active: $a_t \in \{0, 1\}^d$.
        Construct distribution $v_t \in \Delta_d$ as: $v_{t,i} = \frac{a_{t,i} w_{t,i}}{\langle a_t, w_t \rangle}$ for $i \in [d]$.
        Receive loss vector $\ell_t \in \mathbb{R}^d$.
        Construct surrogate loss vector $\widetilde{\ell}_t$ as $\widetilde{\ell}_{t,i} = a_{t,i}(\ell_{t,i} - \langle \ell_t, v_t \rangle)$ for $i \in [d]$, and send it to $\mathcal{A}$.
    **end for**

---

To obtain sleeping experts regret bounds scaling with the norm of the losses, we can implement this reduction with the scale-free Algorithm 4 at its base. Formally, we have the following statement.

**Theorem D.4** (Scale-Free Sleeping Experts Algorithm). *Consider a sleeping experts setting with $d$ experts. Initialize Algorithm 5 using Algorithm 4 (an instance of SOLO FTRL with settings described in Corollary D.3) as its base OLO subroutine. Call the resulting sleeping experts algorithm $\mathcal{A}_{SOLO\,SE}$, with the pseudocode given in Algorithm 6. Then, SOLO SE obtains the following sleeping experts regret bound on any sequence of losses $\{\ell_t\}_{t \in [T]}$:*

$$\text{RegSE}_T\left(\mathcal{A}_{SOLO\,SE}\left(\{\ell_t\}_{t \in [T]}\right)\right) \leq 15 \max_{t \in [T]}\|\ell_t\|_\infty \sqrt{dT}.$$

### D.3. First-Order Neyman Regret Minimization

We now formalize (and generalize) how the ClipOGD design operates. This formalization will define the scope of noncontextual adaptive designs that can be used to estimate group propensities for all groups in our multigroup design.

**Definition D.5** (First-order Neyman Regret Minimization). Recall the Neyman objectives: $f_t(p) = \frac{y_t(1)^2}{p} + \frac{y_t(0)^2}{1-p}$ for $p \in (0, 1), t \geq 1$, where $.\{(y_t(1), y_t(0))\}_{t \geq 1}$ are the potential outcomes.

A first-order Neyman regret minimization algorithm $\mathcal{A}_{\text{ATE}}$ follows the following protocol for sequential ATE estimation: At each round $t = 1, 2, \ldots, \mathcal{A}_{\text{ATE}}$ decides on a treatment probability $p_t \in (1/h(t), 1 - 1/h(t))$, where $h : \mathbb{N}_+ \rightarrow \mathbb{R}_{>0}$ is a

---

**Algorithm 6** $\mathcal{A}_{\text{SOLO SE}}$: Sleeping Experts Algorithm

---

1: Initialize $\mathcal{A}_{SOLO}$, an instance of Algorithm 4.
2: **for** $t = 1, 2, \ldots$ **do**
3:     Receive unscaled weights $w_t \in \mathbb{R}^d_{\geq 0}$ from $\mathcal{A}_{SOLO}$.
4:     Receive indicator vector describing which experts are active: $a_t \in \{0, 1\}^d$.
5:     Set rescaled weights $v_t \in \Delta_d$ as: $v_{t,i} = \frac{a_{t,i} w_{t,i}}{\langle a_t, w_t \rangle}$ for $i \in [d]$.
6:     Receive loss vector $\ell_t \in \mathbb{R}^d$.
7:     Set surrogate loss vector $\widetilde{\ell}_t$ as $\widetilde{\ell}_{t,i} = a_{t,i}(\ell_{t,i} - \langle \ell_t, v_t \rangle)$ for $i \in [d]$.
8:     Send $\widetilde{\ell}_t$ to $\mathcal{A}_{SOLO}$.
9: **end for**

---

strictly increasing clipping function. After that, $\mathcal{A}_{\text{ATE}}$ receives *first-order feedback* $\widetilde{g}_t$ from the environment, which is a random variable that satisfies the following properties: (1) It is adapted to the natural filtration $\{\mathcal{F}_t\}_{t \geq 1}$ of the process, i.e., the distribution of $\widetilde{g}_t$ is determined by all prior history up to and including determining $p_t$; (2) It is an unbiased estimator of $f'_t(p_t)$, in that $\mathbb{E}[\widetilde{g}_t | \mathcal{F}_{t-1}] = f'_t(p_t) = -\frac{y_t(1)^2}{p_t^2} + \frac{y_t(0)^2}{(1-p_t)^2}$.

It is easy to observe that Algorithm 1 conforms to Definition D.5. Algorithm 1 is written as requiring direct access to the selected outcome $Y_t$, but this outcome is only used to compute the unbiased gradient estimator $f'_t(p_t)$.

### D.4. Multigroup-Adaptive Design via Sleeping Experts

We are now ready to present a context-aware algorithm for online ATE estimation. It uses scale-free sleeping experts as derived above, as well first-order Neyman regret minimization algorithms as base learners. The following theorem states its most general guarantees (as well as the specific instantiation that gives MGATE). The proof is presented in the next subsection.

**Theorem D.6** (Guarantees for Algorithm 7)**.** *Consider any first-order Neyman regret minimization algorithm $\mathcal{A}_{\text{ATE}}$ and any scale-free sleeping experts algorithm $\mathcal{A}_{\text{SE}}$. Fix any context space $\mathcal{X}$ and any finite group family $\mathcal{G} \subseteq 2^{\mathcal{X}}$. If the base learners for all $G \in \mathcal{G}$ are copies of $\mathcal{A}_{\text{ATE}}$, Algorithm 7's expected multigroup regret will be bounded for all $G \in \mathcal{G}$ as:*

$$\mathbb{E}\left[\text{RegVar}_T(\mathcal{A}; G)\right] \leq \mathbb{E}\left[\text{RegSE}_T(\mathcal{A}_{\text{SE}})\right] + \mathbb{E}\left[\text{RegVar}_T(\mathcal{A}_{\text{ATE}}(G))\right].$$

*Moreover, Algorithm 7 is* anytime, *as it does not require advance knowledge of the time horizon $T$.*

*Instantiate Algorithm 7 using $h$-clipped ClipOGD$^{\text{SC}}$ as the base ATE algorithm, for some strictly increasing $h$, and use $\mathcal{A}_{\text{SOLO SE}}$ (Algorithm 6) as the scale-free SE algorithm. Then, we obtain the MGATE design (Algorithm 2) that simultaneously offers the following guarantees for all $G \in \mathcal{G}$:*

$$\mathbb{E}\left[\text{RegVar}_T(\mathcal{A}; G)\right] = O\left(\sqrt{|\mathcal{G}|} \cdot (h(T))^5 \cdot \sqrt{T}\right).$$

### D.5. Proof of Theorem D.6

First, note that with the Neyman objective defined, as always, via $f_t(p) = \frac{y_t(1)^2}{p} + \frac{y_t(0)^2}{1-p}$ for $p \in (0, 1)$, we have for any group $G \in \mathcal{G}$:

$$
\begin{aligned}
\text{RegVar}_T(\mathcal{A}; G) &= \sum_{t=1}^{T} \mathbb{1}[x_t \in G]\left(f_t(p_{t,\text{eff}}) - f_t(p^*_{T,G})\right) \\
&= \sum_{t=1}^{T} \mathbb{1}[x_t \in G]\left(f_t\left(\sum_{G' \in \mathcal{G}_t} w_{t,G'} \cdot p_{t,G'}\right) - f_t(p^*_{T,G})\right) \\
&\leq \sum_{t=1}^{T} \mathbb{1}[x_t \in G]\left(\sum_{G' \in \mathcal{G}_t} w_{t,G'} \cdot f_t\left(p_{t,G'}\right) - f_t(p^*_{T,G})\right)
\end{aligned}
$$

---

**Algorithm 7** General Multigroup Adaptive Design

---

**Input:** First-order Neyman regret minimization algorithm $\mathcal{A}_{\text{ATE}}$.
**Input:** Scale-free Sleeping Experts algorithm $\mathcal{A}_{\text{SE}}$.
**Input:** Feature space $\mathcal{X}$, group family $\mathcal{G} \subseteq 2^{\mathcal{X}}$.
Initialize $|\mathcal{G}|$ copies of $\mathcal{A}_{\text{ATE}}$: $\{\mathcal{A}_{\text{ATE}}(G)\}_{G \in \mathcal{G}}$.
**for** $t = 1, 2, \ldots$ **do**
    Get context $x_t \in \mathcal{X}$, let $\mathcal{G}_t = \{G \in \mathcal{G} : x_t \in G\}$.
    **for** active groups $G \in \mathcal{G}_t$ **do**
        Get group-specific advice $p_{t,G}$ from $\mathcal{A}_{\text{ATE}}(G)$.
    **end for**
    Get weights $\{w_{t,G}\}_{G \in \mathcal{G}_t}$ of active groups from $\mathcal{A}_{\text{SE}}$.
    Set treatment probability: $p_{t,\text{eff}} \leftarrow \sum_{G \in \mathcal{G}_t} w_{t,G} \cdot p_{t,G}$.
    Set treatment decision: $Z_t \sim \text{Bernoulli}(p_{t,\text{eff}})$.
    Observe realized outcome: $Y_t \leftarrow y_t(Z_t)$.
    **for** active groups $G \in \mathcal{G}_t$ **do**
        Set estimated loss of $\mathcal{A}_{\text{ATE}}(G)$ as: $\widetilde{\ell}_{t,G} \leftarrow Y_t^2 \left( \frac{Z_t}{p_{t,\text{eff}}} + \frac{1-Z_t}{1-p_{t,\text{eff}}} \right) \left( \frac{Z_t}{p_{t,G}} + \frac{1-Z_t}{1-p_{t,G}} \right)$.
        Set estimated gradient of $\mathcal{A}_{\text{ATE}}(G)$ as: $\widetilde{g}_{t,G} \leftarrow Y_t^2 \left( \frac{Z_t}{p_{t,\text{eff}}} + \frac{1-Z_t}{1-p_{t,\text{eff}}} \right) \left( -\frac{Z_t}{p_{t,G}^2} + \frac{1-Z_t}{(1-p_{t,G})^2} \right)$.
        Send estimated gradient $\widetilde{g}_{t,G}$ back to $\mathcal{A}_{\text{ATE}}(G)$.
    **end for**
    Send estimated losses $\{\widetilde{\ell}_{t,G}\}_{G \in \mathcal{G}_t}$ back to $\mathcal{A}_{\text{SE}}$.
**end for**

---

$$= \underbrace{\sum_{t=1}^{T} \mathbb{1}[x_t \in G] \left( \sum_{G' \in \mathcal{G}_t} w_{t,G'} \cdot f_t(p_{t,G'}) - f_t(p_{t,G}) \right)}_{\text{Term 1: Sleeping Experts Regret of Aggregation Scheme}}$$

$$+ \underbrace{\sum_{t=1}^{T} \mathbb{1}[x_t \in G] \left( f_t(p_{t,G}) - f_t(p_{T,G}^*) \right)}_{\text{Term 2: ATE Neyman regret on Group } G}.$$

Here, $p_{T,G}^*$ denotes the best-in-hindsight static treatment allocation probability on the set of rounds up to round $T$ that correspond to group $G$. The inequality holds by convexity of the objective $f_t$.

What we just did is partition the multigroup regret expression into two terms. The expectation of the second term is bounded by the expected regret of the group-specific ATE Neyman regret minimization algorithm: $\mathbb{E}[\text{Term 2}] \leq \mathbb{E}\left[\text{RegVar}_T(\mathcal{A}_{\text{ATE}})\right]$. The first term will be bounded by the sleeping experts regret of the aggregation algorithm.

To continue the analysis, we first collect the properties of the estimated outcomes, losses, and gradients. Namely, we have for any round $t$ and for any group $G \in \mathcal{G}_t$:

- $\mathbb{E}\left[ \widetilde{\ell}_{t,G} \,\middle|\, \{Z_\tau\}_1^{t-1} \right] = p_{t,\text{eff}} \cdot \frac{(y_t(1))^2}{p_{t,\text{eff}} \cdot p_{t,G}} + (1 - p_{t,\text{eff}}) \cdot \frac{(y_t(0))^2}{(1-p_{t,\text{eff}}) \cdot (1-p_{t,G})} = f_t(p_{t,G})$;

- $\left\| \widetilde{\ell}_t \right\|_\infty = \max_{G \in \mathcal{G}_t} \left| \widetilde{\ell}_{t,G} \right| \leq \max_{G \in \mathcal{G}_t} \max \left\{ \frac{(y_t(1))^2}{p_{t,\text{eff}} \cdot p_{t,G}}, \frac{(y_t(0))^2}{(1-p_{t,\text{eff}}) \cdot (1-p_{t,G})} \right\} \leq C^2 h(t)^2$;

- $\mathbb{E}\left[ \widetilde{g}_{t,G} \,\middle|\, \{Z_\tau\}_1^{t-1} \right] = (1 - p_{t,\text{eff}}) \cdot \frac{(y_t(0))^2}{(1-p_{t,\text{eff}}) \cdot (1-p_{t,G})^2} - p_{t,\text{eff}} \cdot \frac{(y_t(1))^2}{p_{t,\text{eff}} \cdot p_{t,G}^2} = f_t'(p_{t,G})$;

- $\mathbb{E}\left[ |\widetilde{g}_{t,G}| \,\middle|\, \{Z_\tau\}_1^{t-1} \right] = (1 - p_{t,\text{eff}}) \cdot \frac{(y_t(0))^2}{(1-p_{t,\text{eff}}) \cdot (1-p_{t,G})^2} + p_{t,\text{eff}} \cdot \frac{(y_t(1))^2}{p_{t,\text{eff}} \cdot p_{t,G}^2} \leq 2C^2 h(t)^2$;

- $\mathbb{E}\left[ \widetilde{g}_{t,G}^2 \,\middle|\, \{Z_\tau\}_1^{t-1} \right] = (1 - p_{t,\text{eff}}) \cdot \left( \frac{(y_t(0))^2}{(1-p_{t,\text{eff}}) \cdot (1-p_{t,G})^2} \right)^2 + p_{t,\text{eff}} \cdot \left( \frac{(y_t(1))^2}{p_{t,\text{eff}} \cdot p_{t,G}^2} \right)^2 \leq 2C^4 h(t)^5$.

The last calculation holds owing to the fact that at any round $t$, the aggregated probability $p_{t,\text{eff}}$ is a convex combination of the probabilities $p_{t,G}$ for all $G \in \mathcal{G}_t$. Indeed that implies

$$\min\{p_{t,\text{eff}}, (1 - p_{t,\text{eff}})\} \geq \min_{G \in \mathcal{G}_t} \{p_{t,G}, (1 - p_{t,G})\} \geq 1/h(t),$$

leading to the bound $\max\{1/p_{t,\text{eff}}, 1/(1 - p_{t,\text{eff}})\} \leq h(t)$.

By the first of these properties, we can bound the expectation of Term 1 as follows:

$$
\begin{aligned}
\mathbb{E}[\text{Term 1}] &= \mathbb{E}\left[\sum_{t=1}^{T} \mathbb{1}[x_t \in G] \left(\sum_{G' \in \mathcal{G}_t} w_{t,G'} \cdot f_t(p_{t,G'}) - f_t(p_{t,G})\right)\right] \\
&= \mathbb{E}\left[\sum_{t=1}^{T} \mathbb{1}[x_t \in G] \left(\sum_{G' \in \mathcal{G}_t} w_{t,G'} \cdot \mathbb{E}\left[\widetilde{\ell}_{t,G'} \mid \{Z_\tau\}_1^{t-1}\right] - \mathbb{E}\left[\widetilde{\ell}_{t,G} \mid \{Z_\tau\}_1^{t-1}\right]\right)\right] \\
&= \mathbb{E}\left[\sum_{t=1}^{T} \mathbb{1}[x_t \in G] \cdot \mathbb{E}\left[\sum_{G' \in \mathcal{G}_t} w_{t,G'} \cdot \widetilde{\ell}_{t,G'} - \widetilde{\ell}_{t,G} \mid \{Z_\tau\}_1^{t-1}\right]\right] \\
&= \mathbb{E}\left[\sum_{t=1}^{T} \mathbb{1}[x_t \in G] \cdot \mathbb{E}\left[\langle w_t, \widetilde{\ell}_t\rangle - \widetilde{\ell}_{t,G} \mid \{Z_\tau\}_1^{t-1}\right]\right] \\
&= \sum_{t=1}^{T} \mathbb{1}[x_t \in G] \cdot \mathbb{E}\left[\mathbb{E}\left[\langle w_t, \widetilde{\ell}_t\rangle - \widetilde{\ell}_{t,G} \mid \{Z_\tau\}_1^{t-1}\right]\right] \\
&= \mathbb{E}\left[\sum_{t=1}^{T} \mathbb{1}[x_t \in G] \left(\langle w_t, \widetilde{\ell}_t\rangle - \widetilde{\ell}_{t,G}\right)\right] \\
&\leq \mathbb{E}\left[\text{RegSE}_T(\mathcal{A}_{\text{SE}})\right].
\end{aligned}
$$

Thus, in combination with the above we indeed have:

$$\mathbb{E}\left[\text{RegVarMG}_T(\mathcal{A}; \mathcal{G})\right] \leq \mathbb{E}\left[\text{RegSE}_T(\mathcal{A}_{\text{SE}})\right] + \mathbb{E}\left[\text{RegVar}_T(\mathcal{A}_{\text{ATE}})\right].$$

We are now going to instantiate this regret bound with the following concrete choices. $\mathcal{A}_{\text{SE}}$ will be instantiated as the scale-free sleeping experts Algorithm 6. For each copy of the first-order ATE Neyman regret minimization method, we will use the modification of ClipOGD$^{\text{SC}}$ which uses, instead of its originally specified gradient estimator $g_t$, the gradient estimator $\widetilde{g}_t$ specified in our multigroup Algorithm 7.

Our specific choice of $\mathcal{A}_{\text{SE}}$ thus leads, by Theorem D.4 with our above bound on $\left\|\widetilde{\ell}_t\right\|_\infty$ plugged in, to the following bound:

$$\mathbb{E}\left[\text{RegSE}_T(\mathcal{A}_{\text{SE}})\right] \leq 15 \max_{t \in [T]} \|\ell_t\|_\infty \sqrt{dT} \leq 15C^2 \sqrt{|\mathcal{G}|} \cdot (h(T))^2 T^{1/2}.$$

Now we update the regret bound of ClipOGD$^{\text{SC}}$ to use $\widetilde{g}_t$ instead of $g_t$ at each round $t$. From the analysis of Claim 5 and Claim 6 in the proof of Theorem 3.2, we can distill the following inequality holding for *any* unbiased gradient estimators $\{\widetilde{g}_t\}_{t \geq 1}$ and for the optimal $p^*$:

$$\mathbb{E}[\text{RegVar}_T(\mathcal{A}_{\text{ATE}})] = \sum_{t=1}^{T} \mathbb{E}[f_t(p_t) - f_t(p^*)] \leq \sum_{t=1}^{T} \eta_t \cdot \mathbb{E}[\widetilde{g}_t^2 | \mathcal{F}_{t-1}] + 2 \sum_{t=1}^{t^*} \left(\frac{1}{\eta_t \cdot h(t)} + \mathbb{E}[|\widetilde{g}_t| | \mathcal{F}_{t-1}]\right).$$

So it suffices to bound the first and second absolute raw moment of $\widetilde{g}_t$ in terms of the overlap function $h$ in order to obtain concrete regret bounds. From the facts established above, we have at any time horizon $T$ of the multigroup algorithm: $\mathbb{E}\left[\widetilde{g}_{t,G}^2 \mid \{Z_\tau\}_1^{t-1}\right] \leq 2C^4 h(t)^4 h(T)$, and $\mathbb{E}\left[|\widetilde{g}_{t,G}| \mid \{Z_\tau\}_1^{t-1}\right] \leq 2C^2 h(t) h(T)$. Plugging this in and recalling the learning rate setting $\eta_t = \frac{1}{2c^2 t}$, we obtain:

$$\mathbb{E}[\text{RegVar}_T(\mathcal{A}_{\text{ATE}})] \leq \frac{(Ch(T))^5}{2c^2}(\log(T+1) + 1) + 2t^* \cdot C^2 h(t^*) h(T) + 4c^2 \sum_{t=1}^{t^*} \frac{t}{h(t)}$$

$$\leq \frac{(Ch(T))^5}{2c^2}(\log(T+1)+1) + 2C^2\left(1+\frac{C}{c}\right)h_{\text{inv}}\left(1+\frac{C}{c}\right)\cdot h(T) + 2c^2\left(h_{\text{inv}}\left(1+\frac{C}{c}\right)+1\right)^2$$
$$= O\left((h(T))^5 \log T\right).$$

Thus, asymptotically in $T$ this modification of ClipOGD$^{\text{SC}}$ obtains the same rate. The only difference is non-asymptotic; we now acquire an additional term that depends on $T$ in a low-order way: $2C^2(1+C/c)\,h_{\text{inv}}(1+C/c)\cdot h(T)$, which formerly had an additional factor of $(1+C/c)$ instead of $h(T)$. Even though this term is lower-order in $T$, but nonetheless it merits a mention, as here $h(T)$ coexists with the inverse clipping rate mapping, $h_{\text{inv}}$, being evaluated at a "critical point" $1+C/c$. Since the inverse mapping $h_{\text{inv}}$ will grow fast when $h$ grows slowly, this term can practically speaking become influential in the regret bound if the problem is not well-conditioned (if $C/c$ is very large). Thus, in practice this term may merit a tradeoff in choosing $h$ to not be too slowly-growing.

To conclude the proof, note by collecting the above two bounds that:

$$\mathbb{E}\left[\text{RegVarMG}_T(\mathcal{A};\mathcal{G})\right] \leq \mathbb{E}\left[\text{RegSE}_T(\mathcal{A}_{\text{SE}})\right] + \mathbb{E}\left[\text{RegVar}_T(\mathcal{A}_{\text{ATE}})\right]$$
$$\leq 15C^2\sqrt{|\mathcal{G}|}\cdot(h(T))^2 T^{1/2} + O\left((h(T))^5 \log T\right)$$
$$\leq O\left(\sqrt{|\mathcal{G}|}\cdot(h(T))^5\cdot\sqrt{T}\right)$$

# E. Additional Experimental Results

In this section, we present experiments on additional real-world dataset. We list them below along with their descriptions. We then turn to the description of the results.

## E.1. Task and Dataset Descriptions

**Large Language Model Benchmarking**   We test our methods on (a subset of) large language model (LLM) benchmarking data that was examined by Fogliato et al. (2024), and includes BigBench (Srivastava et al., 2023), MedMCQA (Pal et al., 2022), XCOPA (Ponti et al., 2020), HellaSwag (Zellers et al., 2019), MMLU (Hoffmann et al., 2022), and XNLI (Conneau et al., 2018). Multiple LLMs are compared on these datasets, with each model producing a vector of logits for each data point. These logits are turned into probability vectors. Since each task is supervised, we know the correct answers. We select two models—Mistral-7B-Instruct-v0.2 (Jiang et al., 2023) (treatment) and Google Gemma-2b (Team et al., 2024) (control)—and build two parallel outcome sequences using the model accuracies. More specifically, for each data point, the model's chosen answer is the class with the highest predicted probability, and we define accuracy as 1 if this chosen class matches the correct answer and 0 otherwise.

**ASOS Digital Dataset**   We use the sequential experiments dataset from Liu et al. (2021), gathered by ASOS.com between 2019 and 2020. It has 24,153 rows from 78 online controlled experiments. Each row represents a group of users who arrived during a certain time span and shows the average treatment and control outcomes for those users. Across all experiments, there are 99 different treatments and one control. The dataset tracks 4 consistent metrics; each row focuses on one of these metrics. This structure naturally creates 4 subsets of rows (each with about 6,000 rows). We treat each subset as a separate dataset and feed each of these 4 pairs of treatment and control outcome sequences into ClipOGD$^{\text{SC}}$ and ClipOGD$^0$. This setup keeps outcome definitions consistent within each subset, while mixing different experiments and thus providing a varied environment for evaluating sequential ATE estimation methods.

## E.2. Experimental Results

### E.2.1. LLM BENCHMARKING

Figure 4 shows the experimental results across these six tasks (BigBench–MC, HellaSwag, MedMCQA, MMLU, XCOPA, XNLI). The top row shows that the treatment probabilities of ClipOGD$^0$ (orange) fluctuate more, while the treatment probabilities of ClipOGD$^{\text{SC}}$ (blue) settle closer to a stable value. Although our algorithm's assigned probabilities may initially jump around more because of the more aggressive clipping rate, they also stabilize more quickly. The second row shows the variances and tells a similar story: the variance ClipOGD$^{\text{SC}}$ is smaller and decreases faster compared to that of ClipOGD$^0$. As seen in the bottom row, the Neyman regret of ClipOGD$^0$ stays away from zero, whereas the regret

of ClipOGD$^{SC}$ shrinks toward zero or remains lower throughout. This pattern suggests that ClipOGD$^{SC}$ converges to the Neyman-optimal probabilities with less fluctuation and lower regret than ClipOGD$^{0}$.

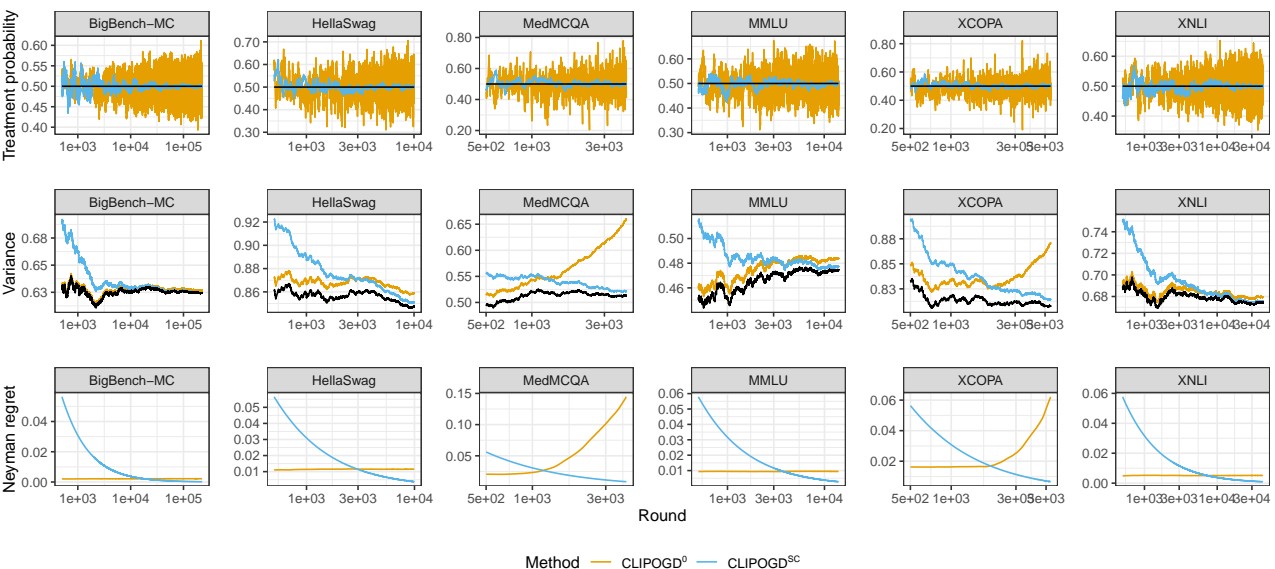

*Figure 4.* **Treatment probabilities, variance of the ATE, and Neyman regret of ClipOGD on LLM benchmarking data**. The solid black line in the treatment probabilities indicates the Neyman optimal probability.

Additionally, we show the per-group Neyman regret of MGATE and ClipOGD in the contextual experiments.

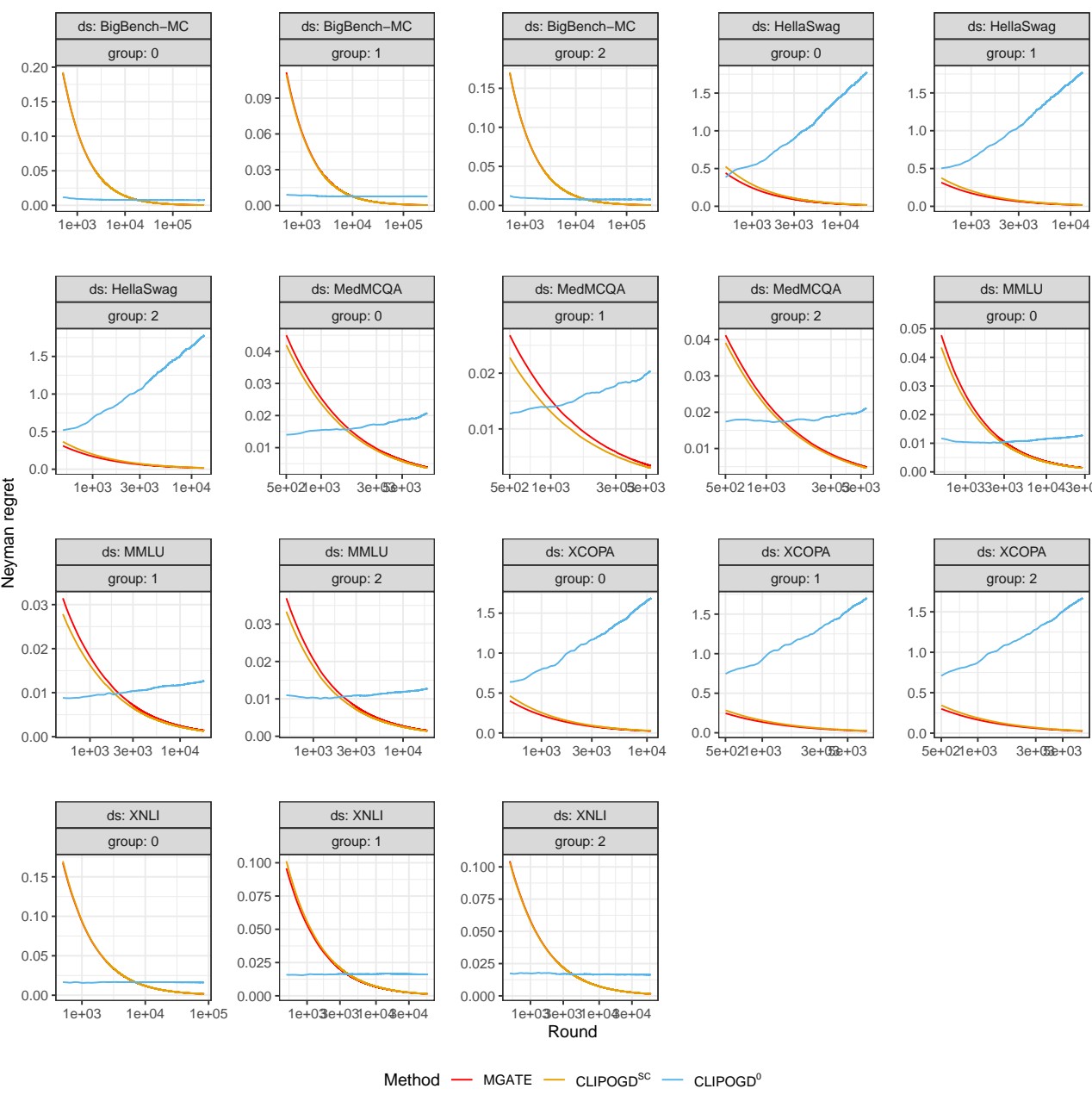

*Figure 5.* **Group-conditional Neyman regret of ClipOGD and MGATE on the LLM Benchmarking data**.

### E.2.2. ASOS DIGITAL DATASET

Figure 6 shows the Neyman regret on this dataset. Across all four metrics, ClipOGD$^{SC}$ (blue) steadily reduces Neyman regret, whereas ClipOGD$^0$ (orange) remains higher or grows over time. Although the regret levels vary by metric, ClipOGD$^{SC}$ consistently converges closer to the Neyman-optimal probabilities as shown by the shrinking regret.

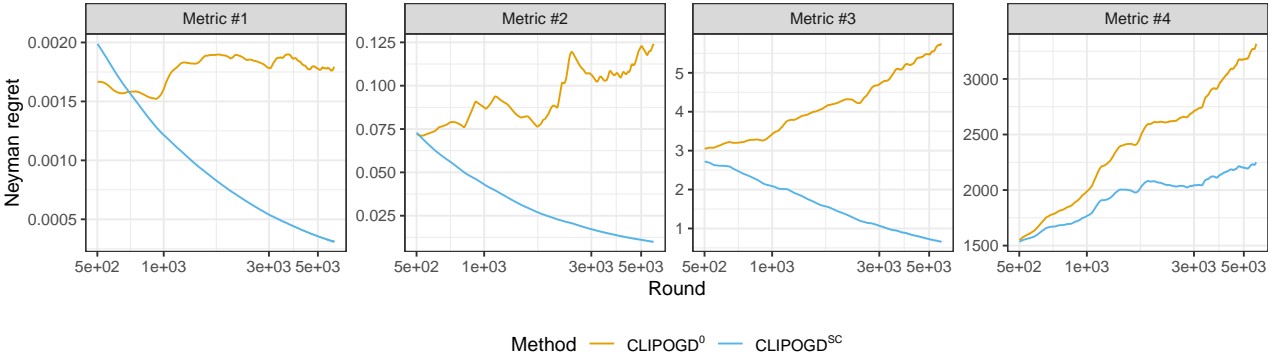

*Figure 6.* **Neyman regret of ClipOGD on the ASOS Digital Dataset.**

Additionally, we show the per-group Neyman regret of MGATE and ClipOGD in the contextual experiments. Here we observe that MGATE and ClipOGD$^{SC}$ attain close to optimal Neyman regret guarantees on all groups.

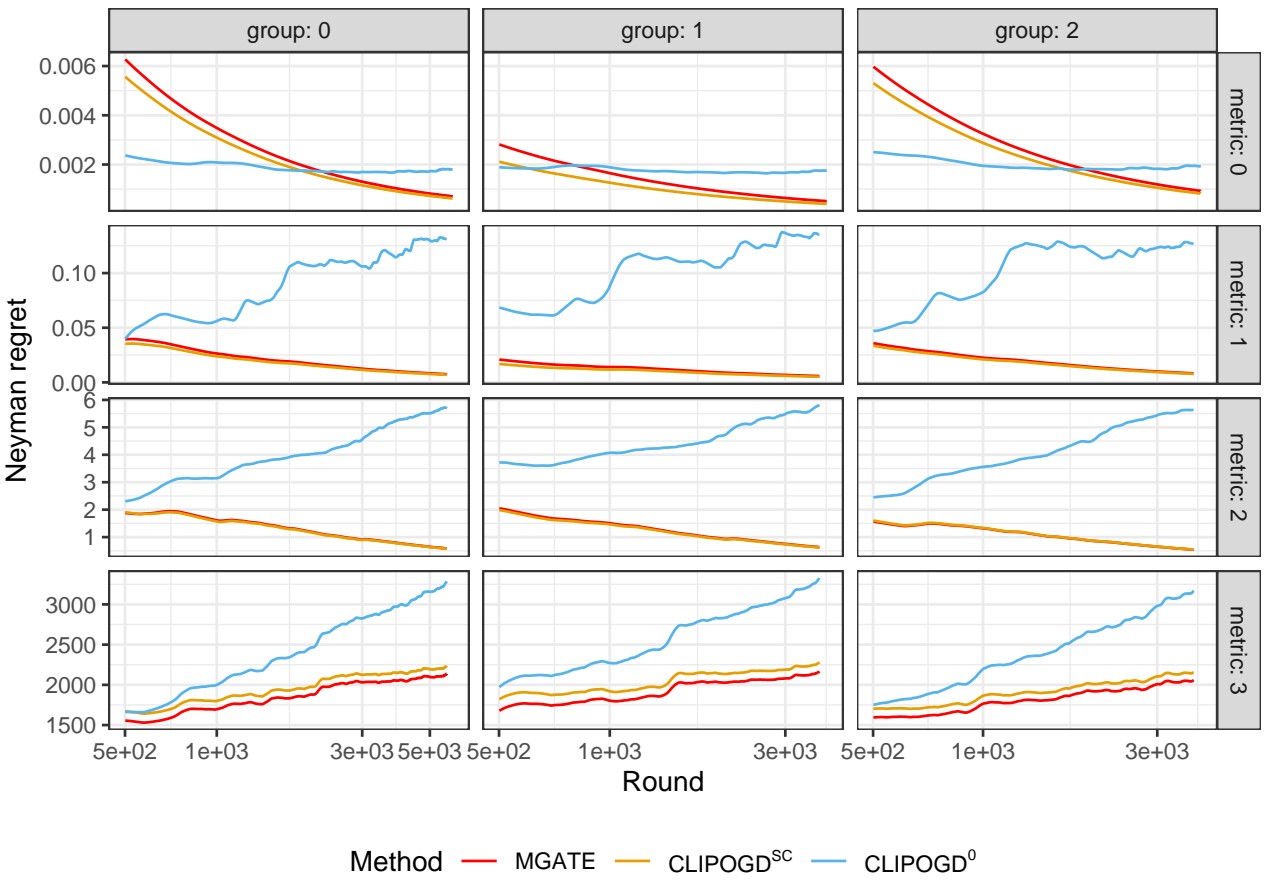

*Figure 7.* **Group-conditional Neyman regret of ClipOGD and AMGATE on the ASOS Digital Dataset**.

