# OpenReview forum: "Stronger Neyman Regret Guarantees for Adaptive Experimental Design"
_ICML.cc/2025/Conference — ICML 2025 spotlightposter_

### Official Review · Reviewer_j79P · 2025-02-28

**Overall Recommendation:** 4

**Summary:**

Based on a stronger assumption, this paper improves the vanilla Neyman regret upper bound from Dai et al., 2023 by modifying the parameters of an existing algorithm. Additionally, the paper considers a contextual multi-group Neyman regret upper bound, for which the authors propose a corresponding algorithm and prove that it achieves a $\mathcal{O}(\sqrt{T})$ regret.

**Claims And Evidence:**

Yes

**Essential References Not Discussed:**

N/A

**Experimental Designs Or Analyses:**

Yes

**Methods And Evaluation Criteria:**

Yes

**Other Comments Or Suggestions:**

Building upon Dai et al., 2023, this paper explores various settings and makes substantial theoretical contributions. I am inclined to accept it.
## update after rebuttal: The authors have addressed my concerns, so I recommend acceptance.

**Other Strengths And Weaknesses:**

Strengths:
1. The paper is well structured. The introduction clearly outlines the motivation of the paper and its main theoretical contributions (i.e., improved vanilla Neyman regret upper bound and new contextual Neyman regret upper bound). Additionally, every symbol in the paper is well-defined.
2. The paper makes multiple theoretical contributions. In addition to studying the upper bound of Neyman regret, it also investigates valid confidence intervals and the convergence rate of the best policy.
3. The paper provides a thorough discussion of related work, especially regarding the technical foundations of its main theorem. The authors cite a series of relevant papers and present a clear outline of their proof.
4. The experimental results are comprehensive.

Weaknesses:
1. The paper does not seem to mention the lower bound for contextual Neyman regret. As a result, it is unclear whether the contextual Neyman regret of the proposed algorithm is optimal.
2. Regarding the vanilla Neyman regret, the proposed method in this paper only modifies the learning rate, which may not have high technical novelty.

**Questions For Authors:**

N/A

**Relation To Broader Scientific Literature:**

N/A

**Theoretical Claims:**

Yes

---

> ### Author Rebuttal · Authors · 2025-04-01
>
> We thank the reviewer for the positive assessment of our paper!
>
> We would like to comment/expand on the point about the lower bounds for the contextual multigroup design: The primary goal of our manuscript on that front is to introduce the multigroup approach to the sequential ATE estimation literature, and to obtain the first sublinear multigroup regret bound. We would like to highlight that even the O(sqrt(T)) multigroup bound that we obtain is substantially non-trivial — due among other things to the unbounded gradients of the loss functions, which with more naive approaches (e.g. by adding noise to smooth them out) could lead to super-sqrt(T) regret bounds (as well as the other technical challenges discussed in Section 4.3).
>
> Lower bounds in multigroup settings are to our knowledge unresolved in the online learning community, and as such present a difficult challenge even beyond sequential ATE estimation, so we therefore consider it fair to leave this challenge to follow-up work. We, however, conjecture that the O(sqrt(T)) multigroup regret that we obtained is in fact a tight minimax rate over all families of groups, and give the following brief discussion below (which we would be happy to include in some form in the revision of our paper if you'd like):
>
> Of course, any lower bound from the non-contextual setting also serves as a lower bound for the multigroup setting (as each group sequence separately must obey such a lower bound). In the non-contextual setting, we were able to match the very recently and independently obtained (Li et al (2024), arXiv preprint arXiv:2410.05552) lower bound of Omega(log T) by leveraging the specific structure (strong convexity) of our objective.
>
> However, as the multigroup algorithm generally requires balancing the competing interests of multiple arbitrary groups at once, the O(log(T)) bound would appear to be too optimistic in this setting. Intuitively, note the following bottleneck that would need to be eliminated to go beyond O(sqrt(T)) rates: while the group-specific sub-learners in the method can obtain O(log(T)) regret on their own groups (by exploiting the strongly convex structure), the regret term that comes from *aggregating* over the sub-learners has O(sqrt(T)) magnitude, thus introducing “overhead” that makes the overall multigroup regret bound O(sqrt(T)) despite the better performance of each group’s algorithm individually.

---

> > ### Comment · Reviewer_j79P · 2025-04-01
> >
> > Thank you for your feedback. All my concerns have been addressed, and I will raise my score to 4 and lean toward accepting the paper.

---

> > > ### Author Response · Authors · 2025-04-09
> > >
> > > We appreciate the raised score, and thank you again for your review!

---

### Official Review · Reviewer_CD6v · 2025-03-14

**Overall Recommendation:** 4

**Summary:**

The authors make two main contributions in this paper: Firstly, in the non-contextual adaptive experimental design case, using stronger assumptions on potential outcome bounds, they change the learning rate and clipping schedules in Dai's ClipOGD algorithm to obtain stronger $O(log T)$ Neyman regret guarantees compared to Dai's $O(\sqrt T)$ Neyman regret guarantees.
Secondly, in the contextual adaptive experimental design case, using a variation of the "sleeping experts" algorithm, christened the MGATE algorithm, they leverage pre-experiment covariates for balancing treatment probabilities to obtain $O(\sqrt T)$ multi-group Neyman regret, a new metric that they themselves define in this paper.
Both algorithms are any-time valid and do not require knowledge of the time horizon $T$.

**Claims And Evidence:**

Theorem 3.2 on non-contextual Neyman regret guarantees, for Algorithm 1, and Theorem 4.2 on multi-group Neyman regret guarantees, for Algorithm 2, appear to be the main theoretical results in this paper, although additional results such as Theorem 3.7 on confidence intervals for Algorithm 1, are also provided.

The authors also provide experimental evidence using one synthetic dataset and one real-world micro-finance dataset in the main body of the paper, to demonstrate the efficacy of algorithms 1 and 2.

**Essential References Not Discussed:**

I couldn't think of any missed essential references.

**Experimental Designs Or Analyses:**

Figures 1 (on the synthetic dataset) and 2 (on the microfinance dataset) clearly demonstrate the superiority of the proposed $ClipOGD^{SC}$ algorithm w.r.t. Dai's $ClipOGD^0$ algorithm, in terms of minimizing non-contextual Neyman regret.
In addition, Fig 3 demonstrates that Algorithm 2 (MGATE) is superior to $ClipOGD^{SC}$ (Algorithm 1) and Dai's $ClipOGD^0$ algorithm, in terms of minimizing group-conditional Neyman regret.

I didn't check the experimental results on additional datasets presented in the supplementary material.

**Methods And Evaluation Criteria:**

The proposed methods and / or evaluation criteria appear to be sound for the problem at hand, as discussed under "Claims And Evidence" above and "Theoretical Claims" / "Experimental Designs Or Analyses" below.

**Other Comments Or Suggestions:**

Does the suggestion regarding generalization to arbitrary (non-necessarily group-based) pre-treatment covariate contextual settings under "Relation To Broader Scientific Literature" seem worthwhile to the authors ?

**Other Strengths And Weaknesses:**

Minor weakness:
Prior to presenting Algorithm 1, the authors define the strictly increasing function $h$ using the natural number domain.
However, in Theorem 3.2., the same function $h$ uses the non-negative real number domain.

**Questions For Authors:**

The author's mention Neyman's classical work on batch design, but do not make any attempt to relate their work to Wald's classical work on sequential hypothesis testing or design. Are connections with Wald's work already discussed in other cited references ?

**Relation To Broader Scientific Literature:**

References such as "Balancing covariates in randomized experiments with the Gram–schmidt Walk design" by Harshaw et al., and "On Distributional Discrepancy for Experimental Design with General Assignment Probabilities" by Rao and Zhang, on non-adaptive designs, use pre-treatment covariates to minimize the variance of the ATE. These references go beyond the group-based contextual setting considered by the authors in Algorithm 2, and may allow it to be further generalized for arbitrary pre-treatment covariate contextual settings.

In view of those known results, given the first contribution, the second contribution in this paper didn't surprise me. If space permits, the authors may wish to cite the above two references on non-adaptive designs.

**Theoretical Claims:**

The theoretical claims in Theorems 3.2 and 4.2 appear to be valid, although I didn't check the proofs in detail.

---

> ### Author Rebuttal · Authors · 2025-04-01
>
> We thank the reviewer for the positive assessment of our paper and for the interesting questions.
>
> The classical work by Wald is indeed the historical underpinning of much of the research in sequential experimental design, particularly on the hypothesis testing side. Our work shares the broad motivation of increasing statistical efficiency through adaptivity, though we do not explicitly deal with some of the notions that Wald focused on such as early stopping. To trace an early mention of the more specific problem of adaptive Neyman allocation in sequential design that we consider in our paper, the work of H. Robbins “Some aspects of the sequential design of experiments” (1952) (which came out shortly after Wald’s seminal treatise) mentions it as an important open problem. These references are mentioned in passing e.g. in [DGH’23]; we will add them to our paper along with brief discussion.
>
> Next, in terms of the definition for h, thank you for pointing out the notational discrepancy. Our arguments in fact don’t require h to be integer-valued, so we revert to the continuous range in the revision.
>
> Next, thank you for sharing the references on covariate balancing / distributional discrepancy based experimental designs. While these are conducted in the non-adaptive setting, they share the objective of optimizing the estimation variance, which we pursue in our work. They also offer a principled way to exploit covariates, by assuming the mapping between covariates and outcomes is linear (and trading off robustness and covariate balance). We will cite and discuss these in the final manuscript.
>
> That being said — based on our reading of these references — there appear to be some important differences between the covariate balancing setting and our multigroup setting, making our contextual setting qualitatively different. (i) Our designs adaptively vary the treatment assignment probabilities to match the performance of the best treatment probability in hindsight. The covariate balancing papers focus on fixing the marginal treatment probabilities (0.5 in the former paper, arbitrary q in the latter) and then obtaining the best treatment assignment vector optimizing the estimation variance (more specifically, the balance-robustness tradeoff). (ii) The covariate balancing work is intimately connected to ridge regression. By contrast, our multigroup design doesn’t assume anything about the nature of the mapping from covariates to features. Instead of balancing covariates over all linear models, it optimizes efficiency for all groups simultaneously, more in the spirit of a “multiobjective” problem with respect to the groups.
>
> So in this way, the two settings, while both are covariate-based and target optimal estimation variance, appear to be quite distinct without one implying the other. But we agree with the reviewer that the covariate balancing approach is worthwhile and potentially fruitful to investigate in the Neyman regret setting that we study — e.g. for obtaining Neyman regret bounds in the spirit of linear bandits.

---

> > ### Comment · Reviewer_CD6v · 2025-04-07
> >
> > I am happy with the author's rebuttal to my review. I am glad they will incorporate some small changes with regard to notation and additional references in response to my comments. I have retained my positive assessment for this paper.

---

> > > ### Author Response · Authors · 2025-04-09
> > >
> > > Thank you again for your careful feedback and for the discussion!

---

### Official Review · Reviewer_zBJq · 2025-03-14

**Overall Recommendation:** 4

**Summary:**

This work studies the design of adaptive, sequential experiments for ATE estimation in the design-based potential outcomes setting. The authors develop adaptive designs without/with covariates to achieve sublinear Neyman regret.

**Claims And Evidence:**

Yes

**Essential References Not Discussed:**

Not aware of any.

**Experimental Designs Or Analyses:**

Yes the experimental designs are valid.

**Methods And Evaluation Criteria:**

Yes

**Other Comments Or Suggestions:**

N/A

**Other Strengths And Weaknesses:**

Strength: very clear writing. And the methods are good technical contributions to the literature.
Weakness:
1. Choice of estimator: the authors use IPW; however, in general Hajek version of IPW is much more robust choice in practice. Is the theory also work generally for Hajek estimator?
2. The vanishing propensity score problem in adaptive experiments is actually studied in the literature. In particular, see Hadad 2019: https://www.pnas.org/doi/10.1073/pnas.2014602118. Is the learning rates here related to some weighting methods introduced in this paper?
3. CI for Clipped Adaptive Design: Also check the Hadad paper - might have some connections.
4. Multi-group extension: has the sense of best subgroup identification + best arm identification. Why are the pretreatment covariates not directly used to enhance efficiency?
5. For simulation, can you also report the bias, variance and CI for the estimators with comparison between Dai approach and the proposed method?

**Questions For Authors:**

N/A

**Relation To Broader Scientific Literature:**

The work contributes to sequential randomization with neyman regret guarantees.

**Theoretical Claims:**

I have checked the proof of the main theorems (3.7 and 4.2) which look correct to me.

---

> ### Author Rebuttal · Authors · 2025-04-01
>
> We thank the reviewer for the positive assessment of our paper, in terms of both its contributions and writing, and for the interesting questions!
> To address your points in the Strengths and Weaknesses section:
> 1. Adapting our theory to the Hajek estimator could indeed be a potential follow-up direction. However, such an adaptation would pose significant technical challenges: its variance, which Neyman regret would aim to track, is nonconvex in the propensity weights due to the normalization involved. As such, our online convex optimization-based machinery is not directly applicable there and would likely require substantial modifications and/or relaxed objectives to obtain Neyman regret results similar to ours. At the same time, if one is not committed to using specifically the Hajek estimator, and is generally interested in double robustness or just in further reducing variance compared to the IPW estimator that we study, we believe our methods could be applied to the augmented IPW estimator with significantly less anticipated modifications, as augmented IPW estimators preserve the nature of the optimization problem with respect to the propensity weights.
>
> 2. In terms of comparing our propensity weight clipping approach to the methods of Hadad et al: We explicitly enforce gradually loosening clippings on our treatment probabilities using a monotonic clipping function (while preserving the form of the adaptive IPW estimator), while Hadad et al, somewhat agnostically to the algorithm that produces the propensity weights, re-weigh the estimator by external weights that partially cancel out the propensity weight blowup. These methods, even though both aim at resolving the challenge of vanishing propensities, thus appear quite distinct in nature.
>
> 3. The Hadad et al paper, whose reweighting strategies accomplish asymptotic normality under some mild assumptions, indeed might serve as a potential lead for designing CLT-based CIs for our methods, as you point out. And in general, since the CI construction presented in Theorem 3.7 is quite conservative and doesn't leverage our much faster Neyman regret rates, improving it is a challenging but important avenue of future work. That being said, the Hadad et al results are restricted to the superpopulation (i.i.d.) setting and appear technically nontrivial to rigorously extend to the finite population setting. Of course, the intuitive / black-box nature of their proposed reweightings could still allow practitioners to use the Hadad et al approach as a heuristic add-on to our adaptive designs.
>
> 4. For the multigroup extension, please note that our method does not aim to identify the best subgroup: instead, it takes as input a given family of groups and optimizes the variance on each of them simultaneously. This makes the multigroup setting different from other existing sequential frameworks for ATE estimation that we are aware of. Within this setting, pretreatment covariates are in fact quite directly used to enhance efficiency: our groups are defined by covariates, and as such our multigroup algorithm takes advantage of any present group-specific homogeneity in the data (for the simplest example, just consider e.g. a finite covariate space X, and define a group for each x \in X; then the multigroup design will compete with the best propensity weight for every covariate). Also, note that for a complementary direct use of covariates, our framework can take advantage of estimating treatment and control outcomes via a model f(x), augmenting the IPW estimators we study into AIPW; however, our focus in this paper is on optimizing the propensities while staying agnostic to the IPW/AIPW distinction.
>
> 5. Please find bias, variance and CI plots at: https://imgur.com/1P9sqbZ

---

> > ### Comment · Reviewer_zBJq · 2025-04-03
> >
> > Thanks for the careful responses.
> >
> > 1. the challenge of applying Hajek-type estimator makes sense to me. Using double robust estimators is indeed an alternative option to consider and I am happy to see such as an extension in the future.
> >
> > 2. the explanation make sense to me. Actually I found it interesting that the goal of regret minimization and stable inference motivate different clipping rates. Hadad et al modify the form of IPW to prevent variance from collapsing while keeping the estimator unbiased; in your case i guess exact unbiasedness might not be necessary for bounding regret.
> >
> > 3. Would love to see more in the future on this potential combination or other ways for better CI construction.
> >
> > 4. Yeah I get the point of using multigroups can greatly enhance efficiency - yet this gives me the feeling that this is a specific strategy for estimating the propensity scores in a covariate dependent way (aka discretizing x). With discrete x, this is indeed proved to be optimal; yet with continuous x, such practice is pretty vague in theory. Correct me if I am misinterpreting the methods/connection here. I am still satisfied with the proposal here as discretizating x is a pretty common strategy by practitioners.
> >
> > 5. Thanks for adding this. The results look good to me.

---

> > > ### Author Response · Authors · 2025-04-09
> > >
> > > Thank you again for your effort in reviewing our manuscript! These are some very good points that we definitely hope future work will explore.

---

### Official Review · Reviewer_Wd4Y · 2025-03-16

**Overall Recommendation:** 4

**Summary:**

This paper explores efficient ATE estimation in adaptive experimental designs. The authors focus on Neyman regret, which quantifies the variance difference between the inverse-propensity-weighted (IPW) estimator under the proposed adaptive design and the best fixed design in hindsight. Prior work (e.g., Dai et al., 2023) established a sublinear $O(\sqrt{T})$ bound on Neyman regret. This paper strengthens that result, achieving an $O(\log T)$ bound under slightly stronger assumptions. The analysis is further extended to contextual (multigroup) settings, introducing a method that ensures $O(\sqrt{T})$ regret across multiple overlapping subpopulations. The approach is validated both theoretically and empirically.

**Claims And Evidence:**

1. **$O(\log T)$ Neyman regret in the noncontextual setting**
   While previous work (Dai et al., 2023) established an $O(\sqrt{T})$ regret bound using the ClipOGD method, this paper demonstrates that, under stronger boundedness assumptions on potential outcomes, the regret can be improved to $O(\log T)$.

2. **Multigroup (contextual) extension**
   The paper introduces MGATE, an adaptive design that leverages pre-treatment covariates to form groups (which may overlap) and jointly optimizes assignment probabilities for each group. The resulting multigroup Neyman regret guarantees a sublinear $O(\sqrt{T})$ bound for all predefined groups simultaneously.

3. **Theoretical analysis and empirical validation**
   The paper presents a rigorous theoretical analysis, establishing regret bounds for both noncontextual and contextual settings. Additionally, it provides extensive empirical validation using synthetic and real-world datasets, demonstrating that the proposed methods consistently outperform or match existing approaches in terms of regret reduction.

**Essential References Not Discussed:**

- I found the following potentially relevant paper recently uploaded on arXiv. It might be helpful to briefly discuss this work in your paper.

Neopane et al. (2025). *Optimistic Algorithms for Adaptive Estimation of the Average Treatment Effect.

**Experimental Designs Or Analyses:**

The paper presents a wide range of experiments and confirm the soundness of their proposed method. While I believe this paper primarily focuses on theoretical contributions and methodological advancements rather than experimental results, the effort put into these experiments is highly commendable.

**Methods And Evaluation Criteria:**

1. The overall approach builds on ClipOGD (online gradient descent with clipping) and **sleeping-experts algorithms**. The incorporation of sleeping-experts algorithms is particularly interesting and innovative.

**Other Comments Or Suggestions:**

- The notation in expressions like $\sum_{t=1}^{T} y_t(1) - y_t(0)$ might be clearer with parentheses. For example, in Definition 2.1, you could write: $\sum_{t=1}^{T}(y_t(1) - y_t(0))$ to emphasize that the subtraction is within the summation.
- Li and Owen is cited as "Harrison H Li and Art B Owen. Double machine learning and design in batch adaptive experiments. arXiv preprint arXiv:2309.15297, 2023." It is now accepted in Journal of Causal Inference.

**Other Strengths And Weaknesses:**

None.

**Questions For Authors:**

None.

**Relation To Broader Scientific Literature:**

This study will have an impact on various fields, including economics and epidemiology.

**Theoretical Claims:**

1. I was personally surprised that the authors achieved a regret bound of $O(\log T)$ instead of $O(\sqrt{T})$. The authors clearly explain and rigorously prove their results.
2. The extension to multigroup settings enhances the method's applicability across various contexts.

---

> ### Author Rebuttal · Authors · 2025-04-01
>
> We thank the reviewer for the careful reading and positive assessment of our paper! We will implement the bullet points in Other Comments and Suggestions.
>
> The suggested paper of Neopane et al (2025) is an independent and concurrent work to ours. We are happy to cite and briefly discuss it. Its setting is closely related to that of our paper, but with the substantial difference that they assume a much milder environment in which outcomes/rewards are generated from some joint distribution with time-stationary means and variances (in contrast to our finite-population setting which makes no such assumptions) — and this milder setting crucially enables them to design an algorithm based on UCB-like optimistic policy tracking approach.

---

### Decision · Program_Chairs · 2025-05-01

**Decision:**

Accept (spotlight poster)

**Comment:**

The goal of this study is to propose an algorithm that achieves sublinear regret for the problem of ATE estimation in sequential experiments. The paper constructs an algorithm that improves upon the existing upper bound of $O(\sqrt{T})$ regret to a logarithmic order $ \widetilde{O}(\log T)$. It also extends the framework to settings with multiple groups and contextual information.

The paper is comprehensive and well-written, with clear and substantive claims. The extensions and experimental support are appropriately included, and the discussion is well-balanced without any significant gaps or redundancies. Reviewers raised questions about the relationship to existing methods, but no critical technical concerns were raised. The authors responded appropriately to the questions, and the reviewers reacted positively to their replies.